# ROBUSTNESS OF UNSUPERVISED REPRESENTATION LEARNING WITHOUT LABELS

## ABSTRACT

Unsupervised representation learning leverages large unlabeled datasets and is competitive with supervised learning. But non-robust encoders may affect downstream task robustness. Recently, robust representation encoders have become of interest. Still, all prior work evaluates robustness using a downstream classification task. Instead, we propose a family of unsupervised robustness measures, which are model- and task-agnostic and label-free. We benchmark state-of-the-art representation encoders and show that none dominates the rest. We offer unsupervised extensions to the FGSM and PGD attacks. When used in adversarial training, they improve most unsupervised robustness measures, including certified robustness. We validate our results against a linear probe and show that, for MOCOv2, adversarial training results in 3 times higher certified accuracy, a 2-fold decrease in impersonation attack success rate and considerable improvements in certified robustness.

## 1 INTRODUCTION

Unsupervised and self-supervised models extract useful representations without requiring labels. They can learn patterns in the data and are competitive with supervised models for image classification by leveraging large unlabeled datasets (He et al., 2020; Chen et al., 2020b;d;c; Zbontar et al., 2021; Chen & He, 2021). Representation encoders do not use task-specific labels and can be employed for various downstream tasks. Such reuse is attractive as large datasets can make them expensive to train.

Therefore, applications are often built on top of public domain representation encoders. However, lack of robustness of the encoder can be propagated to the downstream task. Consider the impersonation attack threat model in Fig. 1. An attacker tries to fool a classifier that uses a representation encoder. The attacker has white-box access to the representation extractor (e.g. an open-source model) but they *do not* have access to the classification model that uses the representations. By optimizing the input to be similar to a benign input, but to have the representation of a different target input, the attacker can fool the classifier. Even if the classifier is private, one can attack the combined system if the public encoder conflates two different concepts onto similar representations. Hence, robustness against such conflation is necessary to perform downstream inference on robust features.

We currently lack ways to evaluate robustness of representation encoders without specializing for a particular task. While prior work has proposed improving the robustness of self-supervised representation learning (Alayrac et al., 2019; Kim et al., 2020; Jiang et al., 2020; Ho & Vasconcelos, 2020; Chen et al., 2020a; Cemgil et al., 2020; Carmon et al., 2020; Gowal et al., 2020; Fan et al., 2021; Nguyen et al., 2022; Kim et al., 2022), they all require *labeled* datasets to evaluate the robustness of the resulting models.

Instead, we offer encoder robustness evaluation *without labels*. This is task-agnostic, in contrast to supervised assessment, as labels are (implicitly) associated with a specific task. Labels can also be incomplete, misleading or stereotyping (Stock & Cisse, 2018; Steed & Caliskan, 2021; Birhane & Prabhu, 2021), and can inadvertently impose biases in the robustness assessment. In this work, we propose measures that do not require labels and methods for unsupervised adversarial training that result in more robust models. To the best of our knowledge, this is the first work on unsupervised robustness evaluation and we make the following contributions to address this problem:

1. Novel representational robustness measures based on clean-adversarial representation divergences, requiring no labels or assumptions about underlying decision boundaries.

Figure 1: Impersonation attack threat model. The attacker has access only to the encoder on which the classifier is built. By attacking the input to have a similar representation to a sample from the target class, the attacker can fool the classifier without requiring any access to it. Cats who successfully impersonate dogs under the MOCOv2 representation encoder and a linear probe classifier are shown.

2. A unifying framework for unsupervised adversarial attacks and training, which generalizes the prior unsupervised adversarial training methods.

3. Evidence that even the most basic unsupervised adversarial attacks in the framework result in more robust models relative to both supervised and unsupervised measures.

4. Probabilistic guarantees on the unsupervised robustness measures based on center smoothing.

## 2 RELATED WORK

**Adversarial robustness of supervised learning**    Deep neural networks can have high accuracy on clean samples while performing poorly under imperceptible perturbations (adversarial examples) (Szegedy et al., 2014; Biggio et al., 2013). Adversarial examples can be viewed as spurious correlations between labels and style (Zhang et al., 2022; Singla & Feizi, 2022) or shortcut solutions (Robinson et al., 2021). Adversarial training, i.e. incorporating adversarial examples in the training process, is a simple and widely used strategy against adversarial attacks (Goodfellow et al., 2015; Madry et al., 2018; Shafahi et al., 2019; Bai et al., 2021).

**Unsupervised representation learning**    Representation learning aims to extract useful features from data. Unsupervised approaches are frequently used to leverage large unlabeled datasets. Siamese networks map similar samples to similar representations (Bromley et al., 1993; Koch et al., 2015), but may collapse to a constant representation. However, Chen & He (2021) showed that a simple stop-grad can prevent such collapse. Contrastive learning was proposed to address the representational collapse by introducing negative samples (Hadsell et al., 2006; Le-Khac et al., 2020). It can benefit from pretext tasks (Xie et al., 2021; Bachman et al., 2019; Tian et al., 2020; Oord et al., 2018; Ozair et al., 2020; McAllester & Stratos, 2020). Some methods that do not need negative samples are VAEs (Kingma & Welling, 2014), generative models (Kingma et al., 2014; Goodfellow et al., 2014; Donahue & Simonyan, 2020), or bootstrapping methods such as BYOL by Grill et al. (2020).

**Robustness of unsupervised representation learning**    Most robustness work has focused on supervised tasks, but there has been recent interest in unsupervised training for representation encoders. Kim et al. (2020), Jiang et al. (2020), Gowal et al. (2020) and Kim et al. (2022) propose generating instance-wise attacks by maximizing a contrastive loss and using them for adversarial training. Fan et al. (2021) complement this by a high-frequency view. Ho & Vasconcelos (2020) suggest attacking batches instead of individual samples. KL-divergence can also be used as a loss (Alayrac et al., 2019) or as a regularizer (Nguyen et al., 2022). Alternatively, a classifier can be trained on a small labeled dataset with adversarial training applied to it (Carmon et al., 2020; Alayrac et al., 2019). For VAEs, Cemgil et al. (2020) generate attacks by maximizing the Wasserstein distance to the clean representations in representation space. Peychev et al. (2022) address robustness from the perspective of individual fairness: they certify that samples close in a feature directions are close in representation space. However, their approach is limited to invertible encoders. While these methods obtain robust unsupervised representation encoders, they all evaluate robustness on a single

supervised classification task. To the best of our knowledge, no prior work has proposed measures for robustness evaluation *without* labels.[1]

# 3 PROBLEM SETTING

Let $f : \mathcal{X} \to \mathcal{R}$ be a differentiable encoder from $\mathcal{X} = [0,1]^n$ to a representation space $\mathcal{R}$. We require $f$ to be white-box: we can query both $f(x)$ and $\frac{df(x)}{dx}$. $\mathcal{R}$ is endowed with a divergence $d(r, r')$, a function that has the non-negativity $(d(r, r') \geq 0, \forall r, r' \in \mathcal{R})$ and identity of indiscernibles $(d(r, r') = 0 \Leftrightarrow r = r')$ properties. This includes metrics on $\mathcal{R}$ and statistical distances, e.g. the KL-divergence. $D$ is a dataset of iid samples from a distribution $\mathcal{D}$ over $\mathcal{X}$.

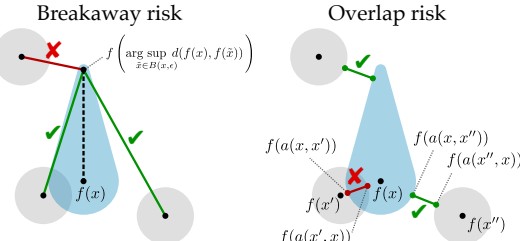

Figure 2: Breakaway and overlap risks. Divergences that increase the corresponding risks are in red and those reducing them are in green.

We denote perturbations of $x$ as $\hat{x} = x + \delta$, $\|\delta\|_\infty \leq \epsilon$, with $\epsilon$ small enough so that $\hat{x}$ is indistinguishable from $x$. We assume to be desirable that $f$ maps $\hat{x}$ close to $x$, i.e. $d(f(x), f(\hat{x}))$ should be "small". We call this property *unsupervised robustness*. It is related to the *smoothness* of encoders (Bengio et al., 2013). Intuitively, (Lipschitz) continuous downstream tasks would translate such unsupervised robustness to supervised robustness.

We propose two risk measures for unsupervised robustness. First, the *breakaway risk* is the probability that the worst-case perturbation of $x$ in the ball $B(x, \epsilon)$ is closer to a different sample $x'$ than to $x$:

$$\mathop{\mathbb{P}}_{x,x'\sim\mathcal{D}} \left[ d(f(\hat{x}), f(x')) < d(f(\hat{x}), f(x)) \right], \quad \hat{x} \in \arg \sup_{\tilde{x} \in B(x,\epsilon)} d(f(x), f(\tilde{x})). \quad (1)$$

Another indication of the lack of unsupervised robustness is if $f(B(x, \epsilon))$ and $f(B(x', \epsilon))$ overlap, as then there exist perturbations $\delta, \delta'$ under which downstream tasks would not be able to distinguish between the two samples, i.e. if $f(x + \delta) = f(x' + \delta')$. We call this the *overlap risk* and define it as:

$$\mathop{\mathbb{P}}_{x,x'\sim\mathcal{D}} \left[ d(f(x), f(a(x', x))) < d(f(x), f(a(x, x'))) \right], \quad a(o, t) \in \arg \inf_{\tilde{x} \in B(o,\epsilon)} d(f(t), f(\tilde{x})). \quad (2)$$

The breakaway risk is based on the perturbation causing the largest divergence in $\mathcal{R}$, while the overlap risk measures if perturbations are well-separated from other instances (see Fig. 2). Labels are not required: the two risks are defined with respect to the local properties of the representation manifold of $f$ under $\mathcal{D}$ and not relative to a task. In fact, we neither explicitly nor implicitly consider decision boundaries or underlying classes and make no assumptions about the downstream task.

What if $x$ and $x'$ are very similar? Perhaps we shouldn't consider breakaway and overlap in such cases? We argue against this. First, the probability of very similar pairs would be low in a sufficiently diverse distribution $\mathcal{D}$. Second, there is no clear notion of similarity on $\mathcal{X}$ without making assumptions about the downstream tasks. Finally, even if $x$ is very similar to $x'$, it should still be more similar to any $\hat{x}$ as $x$ and $\hat{x}$ are defined to be visually indistinguishable. We call this the *self-similarity assumption*.

# 4 UNSUPERVISED ADVERSARIAL ATTACKS ON REPRESENTATION ENCODERS

It is not tractable to compute the supremum and infimum in Eqs. (1) and (2) exactly for a general $f$. Instead, we can approximate them via constrained optimization in the form of adversarial attacks. This section shows how to modify the FGSM and PGD supervised attacks for these objectives (Secs. 4.1 and 4.2), as well as how to generalize these methods to arbitrary loss functions (Sec. 4.3). The adversarial examples can also be used for adversarial training (Sec. 4.4).

---

[1]Concurrent work by Wang & Liu (2022) proposed RVCL: a method to evaluate robustness without labels. However, it focuses on contrastive learning models while the methods here work with arbitrary encoders.

### 4.1 UNSUPERVISED FAST GRADIENT SIGN METHOD (U-FGSM) ATTACK

The Fast Gradient Sign Method (FGSM) is a popular one-step method to generate adversarial examples in the supervised setting (Goodfellow et al., 2015). Its untargeted mode perturbs the input $x$ by taking a step of size $\alpha \in \mathbb{R}_{>0}$ in the direction of maximizing the classification loss $\mathcal{L}_{\text{cl}}$ relative to the true label $y$. In targeted mode, it minimizes the loss of classifying $x$ as a target class $t \neq y$:

$$\hat{x} = \text{clip}(x + \alpha \, \text{sign}(\nabla_x \mathcal{L}_{\text{cl}}(f(x), y))) \qquad \text{untargeted FGSM},$$
$$\hat{x}^{\to t} = \text{clip}(x - \alpha \, \text{sign}(\nabla_x \mathcal{L}_{\text{cl}}(f(x), t))) \qquad \text{targeted FGSM},$$

where $\text{clip}(x)$ clips all values of $x$ to be between 0 and 1.

**Untargeted U-FGSM** We can approximate the supremum in Eq. (1) by replacing $\mathcal{L}_{\text{cl}}$ with the representation divergence $d$, using a small perturbation $\eta \in \mathbb{R}^n$ to ensure non-zero gradient:

$$\hat{x} = \text{clip}(x + \alpha \, \text{sign}(\nabla_x d(f(x), f(x + \eta)))).$$

Ho & Vasconcelos (2020) also propose an untargeted FGSM attack for the unsupervised setting, which requires batches rather than single images and uses a specific loss function, and hence is an instance of the $\mathcal{L}$-FGSM attack in Sec. 4.3. The untargeted U-FGSM proposed here is independent of the loss used for training, making it more versatile.

**Targeted U-FGSM** We can also perturb $x_i \in D$ so that its representation becomes close to $f(x_j)$ for some $x_j \in D, x_j \neq x_i$. Then a downstream model would struggle to distinguish between the attacked input and the target $x_j$. This approximates the infimum in Eq. (2):

$$\hat{x}_i^{\to j} = \text{clip}(x_i - \alpha \, \text{sign}(\nabla_{x_i} d(f(x_i), f(x_j)))).$$

### 4.2 UNSUPERVISED PROJECTED GRADIENT DESCENT (U-PGD) ATTACK

PGD attack is the gold standard of supervised adversarial attacks (Madry et al., 2018). It comprises iterating FGSM and projections onto $B(x, \epsilon)$, the $\ell_\infty$ ball of radius $\epsilon$ centered at $x$:

$$\hat{x}_0 = \hat{x}_0^{\to t} = \text{clip}\left(x + U^n[-\epsilon, \epsilon]\right) \qquad \text{randomized initialization},$$
$$\hat{x}_{u+1} = \text{clip}(\Pi_{x,\epsilon}[\hat{x}_u + \alpha \, \text{sign}(\nabla_{\hat{x}_u} \mathcal{L}_{\text{cl}}(f(\hat{x}_u), y))]) \qquad \text{untargeted PGD},$$
$$\hat{x}_{u+1}^{\to t} = \text{clip}(\Pi_{x,\epsilon}[\hat{x}_u^{\to t} - \alpha \, \text{sign}(\nabla_{\hat{x}_u^{\to t}} \mathcal{L}_{\text{cl}}(f(\hat{x}_u^{\to t}), t))]) \qquad \text{targeted PGD}.$$

We can construct the unsupervised PGD (U-PGD) attacks similarly to the U-FGSM attack:

$$\hat{x}_0 = \hat{x}_0^{\to t} = \text{clip}\left(x + U^n[-\epsilon, \epsilon]\right) \qquad \text{randomized initialization},$$
$$\hat{x}_{u+1} = \text{clip}(\Pi_{x,\epsilon}[\hat{x}_u + \alpha \, \text{sign}(\nabla_{\hat{x}_u} d(f(\hat{x}_u), f(x)))]) \qquad \text{untargeted U-PGD},$$
$$\hat{x}_{u+1}^{\to t} = \text{clip}(\Pi_{x,\epsilon}[\hat{x}_u^{\to t} - \alpha \, \text{sign}(\nabla_{\hat{x}_u^{\to t}} d(f(\hat{x}_u^{\to t}), f(x_j)))]) \qquad \text{targeted U-PGD}.$$

By replacing the randomized initialization with the $\eta$ perturbation in the first iteration of the untargeted case, one obtains an unsupervised version of the BIM attack (Kurakin et al., 2017). The adversarial training methods proposed by Alayrac et al. (2019); Cemgil et al. (2020); Nguyen et al. (2022) can be considered as using U-PGD attacks with specific divergence choices (see App. A.1).

### 4.3 LOSS-BASED ATTACKS

In both their supervised and unsupervised variants, FGSM and PGD attacks work by perturbing the input in order to maximize or minimize the divergence $d$. By considering arbitrary differentiable loss functions instead, we can define a more general class of *loss-based attacks*.

**Instance-wise loss-based attacks ($\mathcal{L}$-FGSM, $\mathcal{L}$-PGD)** Given an instance $x \in \mathcal{X}$ and a loss function $\mathcal{L} : (\mathcal{X} \to \mathcal{R}) \times \mathcal{X} \to \mathbb{R}$, the $\mathcal{L}$-FGSM attack takes a step in the direction maximizing $\mathcal{L}$:

$$\hat{x} = \text{clip}\left(x + \alpha \, \text{sign}\left(\nabla_x \mathcal{L}(f, x)\right)\right).$$

Similarly, for a loss function $\mathcal{L} : (\mathcal{X} \to \mathcal{R}) \times \mathcal{X} \times \mathcal{X} \to \mathbb{R}$ taking a representation encoder, a sample, and the previous iteration of the attack, the loss-based PGD attack is defined as:

$$\hat{x}_0 = \text{clip}\left(x + U^n[-\epsilon, \epsilon]\right),$$
$$\hat{x}_{u+1} = \text{clip}\left(\Pi_{x,\epsilon}\left[\hat{x}_u + \alpha \, \text{sign}(\nabla_{\hat{x}_u} \mathcal{L}(f, x, \hat{x}_u)]\right]\right).$$

If we do not use random initialization for the attack, we get the $\mathcal{L}$-BIM attack.

The supervised and unsupervised FGSM and PGD attacks are special cases of the $\mathcal{L}$-FGSM and $\mathcal{L}$-PGD attacks. Furthermore, prior unsupervised adversarial training methods can also be represented as $\mathcal{L}$-PGD attacks (Kim et al., 2020; Jiang et al., 2020). A full description is provided in App. A.1.

**Batch-wise loss-based attacks ($\bar{\mathcal{L}}$-FGSM, $\bar{\mathcal{L}}$-PGD)**  Attacking whole batches instead of single inputs can account for interactions between the individual inputs in a batch. The above attacks can be naturally extended to work over batches by independently attacking all inputs from $X = [x_1, \ldots, x_N]$. This can be done with a more general loss function $\bar{\mathcal{L}} : (\mathcal{X} \to \mathcal{R}) \times \mathcal{X}^N \to \mathbb{R}$. The batch-wise loss-based FGSM attack $\bar{\mathcal{L}}$-FGSM is provided in Eq. (3) with $\bar{\mathcal{L}}$-PGD and $\bar{\mathcal{L}}$-BIM defined similarly.

$$\hat{X} = \text{clip}\left(X + \alpha \, \text{sign}\left(\nabla_X \bar{\mathcal{L}}(f, X)\right)\right). \tag{3}$$

Any instance-wise loss-based attack can be trivially represented as a batch-wise attack. Additionally, prior unsupervised adversarial training methods can also be represented as $\bar{\mathcal{L}}$-FGSM and $\bar{\mathcal{L}}$-PGD attacks (Ho & Vasconcelos, 2020; Fan et al., 2021; Jiang et al., 2020) (see App. A.2).

### 4.4 ADVERSARIAL TRAINING FOR UNSUPERVISED LEARNING

Adversarial training is a min-max problem minimizing a loss relative to a worst-case perturbation that maximizes it (Goodfellow et al., 2015). As the worst-case perturbation cannot be computed exactly (similarly to Eqs. (1) and (2)), adversarial attacks are usually used to approximate it. Any of the aforementioned attacks can be used for the inner optimization for adversarial training. Prior works use divergence-based (Alayrac et al., 2019; Cemgil et al., 2020; Nguyen et al., 2022) and loss-based attacks (Kim et al., 2020; Jiang et al., 2020; Ho & Vasconcelos, 2020; Fan et al., 2021). These methods tend to depend on complex loss functions and might work only for certain models. Therefore, we propose using targeted or untargeted U-PGD, as well as $\bar{\mathcal{L}}$-PGD with the loss used for training. They are simple to implement and can be applied to any representation learning model.

## 5 ROBUSTNESS ASSESSMENT WITH NO LABELS

The success of a supervised attack is clear-cut: whether the predicted class is different from the one of the clean sample. In the unsupervised case, however, it is not clear when an adversarial attack results in a representation that is "too far" from the clean one. In this section, we propose using quantiles to quantify distances and discuss estimating the breakaway and overlap risks (Eqs. (1) and (2)).

**Universal quantiles (UQ) for untargeted attacks**  For untargeted attacks, we propose measuring $d(f(\hat{x}), f(x))$ relative to the distribution of divergences between representations of samples from $\mathcal{D}$. In particular, we suggest reporting the quantile $q = \mathbb{P}_{x',x'' \sim \mathcal{D}}\left[d(f(x'), f(x'')) \leq d(f(\hat{x}), f(x))\right]$. This measure is independent of downstream tasks and depends only on the properties of the encoder and $\mathcal{D}$. We can use it to compare different models, as it is agnostic to the different representation magnitudes models may have. In practice, the quantile values can be estimated from the dataset $D$.

**Relative quantiles (RQ) for targeted attacks**  Nothing prevents universal quantiles to be applied to targeted attacks. However, considering that targeted attacks try to "impersonate" a particular target sample, we propose using *relative quantiles* to assess their success. We assess the attack as the distance $d(f(\hat{x}_i^{\to j}), f(x_j))$ induced by the attack relative to $d(f(x_i), f(x_j))$, the original distance between the clean sample and the target. The relative quantile for a targeted attack $\hat{x}_i^{\to j}$ is then the ratio $d(f(\hat{x}_i^{\to j}), f(x_j))/d(f(x_i), f(x_j))$.

Quantiles are a good way to assess the success of individual attacks or to compare different models. However, they do not take into account the local properties of the representation manifold, i.e. that

some regions of $\mathcal{R}$ might be more densely populated than others. The breakaway and overlap risk metrics were defined with this exact purpose. Hence, we propose estimating them.

**Estimating the breakaway risk**  While the supremum in Eq. (1) cannot be computed explicitly, it can be approximated using the untargeted U-FGSM and U-PGD attacks. Therefore, we can compute a Monte Carlo estimate of Eq. (1) by sampling pairs $(x, x')$ from the dataset $D$ and performing an untargeted attack on $x$, for example with U-PGD.

**Nearest neighbour accuracy**  As the breakaway risk can be very small for robust encoders we propose also reporting the fraction of samples in $D' \subseteq D$ whose untargeted attacks $\hat{x}$ would have their nearest clean neighbour in $D$ being their corresponding clean samples $x$. That is:

$$\frac{1}{|D'|} \sum_{x \in D'} \mathbb{1} \left[ \nexists x' \in D, x' \neq x, \text{ s.t. } d(f(x'), f(\hat{x})) < d(f(x), f(\hat{x})) \right]. \tag{4}$$

**Estimating the overlap risk**  The infimums in Eq. (2) can be estimated with an unsupervised targeted attack. Hence, an estimate of Eq. (2) can be computed by sampling pairs $(x_i, x_j)$ from the dataset $D$ and computing the targeted attacks $\hat{x}_i^{\rightarrow j}$ and $\hat{x}_j^{\rightarrow i}$. The overlap risk estimate is then the fraction of pairs for which $d(f(x_i), f(\hat{x}_j^{\rightarrow i})) < d(f(x_i), f(\hat{x}_i^{\rightarrow j}))$.

**Adversarial margin**  In Eq. (2) one takes into account only whether overlap occurs but not the magnitude of the violation. Therefore, we also propose looking at the margin between the two attacked representations, normalized by the divergence between the clean samples:

$$\frac{d(f(x_i), f(\hat{x}_j^{\rightarrow i})) - d(f(x_i), f(\hat{x}_i^{\rightarrow j}))}{d(f(x_i), f(x_j))},$$

for randomly selected pairs $(x_i, x_j)$ from $D$. If overlap occurs, this ratio would be negative, with more negative values pointing to stronger violations. The overlap risk is therefore equivalent to the probability of occurrence of a negative adversarial margin.

**Certified unsupervised robustness**  The present work depends on gradient-based attacks, which can be fooled by gradient masking (Athalye et al., 2018; Uesato et al., 2018). Hence, we also assess the certified robustness of the encoder. By using center smoothing (Kumar & Goldstein, 2021) we can compute a probabilistic guarantee on the radius of the $\ell_2$-ball in $\mathcal{R}$ that contains at least half of the probability mass of $f(x + \mathcal{N}(0, \sigma^2))$. The smaller this radius is, the closer $f$ maps similar inputs. Hence, this is a probabilistically certified alternative to assessing robustness via untargeted attacks. In order to compare certified radius values in $\mathcal{R}$ across models we report them as universal quantiles.

# 6 EXPERIMENTS

We assess the robustness of state-of-the-art representation encoders against the unsupervised attacks and robustness measures from Secs. 4 and 5. We consider the ResNet50-based self-supervised models MOCOv2 (He et al., 2020; Chen et al., 2020d), MOCO with non-semantic negatives (Ge et al., 2021), PixPro (Xie et al., 2021), AMDIM (Bachman et al., 2019), SimCLRv2 (Chen et al., 2020c), and SimSiam (Chen & He, 2021). To compare self-supervised and supervised methods, we also evaluate the penultimate layer of ResNet50 (He et al., 2016) and supervised adversarially trained ResNet50 (Salman et al., 2020). We assess the effect of using different unsupervised attacks by fine-tuning MOCOv2 with the untargeted U-PGD, targeted U-PGD, as well as with $\bar{\mathcal{L}}$-PGD using MOCOv2's contrastive loss, as proposed in Sec. 4.4. App. B reports the performance of unsupervised adversarial training on ResNet50 and the transformer-based MOCOv3 (Chen et al., 2021).

The unsupervised evaluation uses the PASS dataset as it does not contain people and identifiable information and has proper licensing (Asano et al., 2021). ImageNet (Russakovsky et al., 2015) is used for accuracy benchmarking and the adversarial fine-tuning of MOCO, as to be consistent with how the model was trained. Assira (Elson et al., 2007) is used for the impersonation attacks.

We report median UQ and RQ for the $\ell_2$ divergence for the untargeted and targeted U-PGD attacks with $\epsilon = 0.05$ and $\epsilon = 0.10$. We also estimate the breakaway risk, nearest neighbour accuracy, overlap risk and adversarial margin, and certified unsupervised robustness, as described in Sec. 5.

Table 1: Standard and lowpass ImageNet accuracy of linear probes of ResNet50-based encoders.

| Accuracy | ResNet50 | Robust ResNet50 | Standard ResNet-based unsupervised models | | | | | | Adversarially fine-tuned MOCOv2 | | |
|---|---|---|---|---|---|---|---|---|---|---|---|
| | | | PixPro | AMDIM | SimCLR | SimSiam | Nonsem | MOCOv2 | TAR | UNTAR | $\overline{\mathcal{L}}$−PGD |
| Standard | 74% | 62% | 58% | 62% | 67% | 63% | 50% | 67% | 60% | 60% | 57% |
| Lowpass | 68% | 59% | 50% | 53% | 60% | 56% | 47% | 62% | 59% | 58% | 55% |

Table 2: Robustness of ResNet50 and ResNet50-based unsupervised encoders on PASS (except the average certified radius measured on ImageNet). Arrows show if larger or smaller values are better.

| | | | ResNet50 | Robust ResNet | Standard ResNet-based unsupervised models | | | | | |
|---|---|---|---|---|---|---|---|---|---|---|
| | | | | | PixPro | AMDIM | SimCLR | SimSiam | Nonsem | MOCOv2 |
| Targeted U-PGD | $\varepsilon = 0.05$ | 5 iter. (RQ) ↑ | 66.38% | 98.14% | **80.51%** | **80.82%** | 75.04% | 73.25% | 61.54% | 65.64% |
| | | 10 iter. (RQ) ↑ | 52.68% | 96.45% | 67.91% | **70.26%** | 65.77% | 65.94% | 46.52% | 52.08% |
| | | 50 iter. (RQ) ↑ | 27.42% | 86.89% | 37.57% | **44.28%** | 36.38% | 28.13% | 22.23% | 20.51% |
| | $\varepsilon = 0.10$ | 5 iter. (RQ) ↑ | 69.27% | 97.92% | 83.06% | **85.59%** | 78.02% | 78.11% | 64.28% | 68.68% |
| | | 10 iter. (RQ) ↑ | 55.63% | 96.03% | 71.18% | **75.93%** | 69.01% | 70.45% | 48.57% | 55.02% |
| | | 50 iter. (RQ) ↑ | 27.92% | 83.90% | 38.07% | **47.28%** | 38.60% | 30.33% | 22.32% | 21.88% |
| Untar. U-PGD | $\varepsilon = 0.05$ | 5 iter. (UQ) ↓ | 98.70% | 0.00% | **14.35%** | 81.30% | 43.65% | 72.55% | 46.80% | 65.10% |
| | | 10 iter. (UQ) ↓ | 99.90% | 0.00% | **85.60%** | 98.50% | 99.40% | 98.60% | 98.10% | 99.00% |
| | | 50 iter. (UQ) ↓ | 99.90% | 0.01% | 99.90% | 99.90% | 99.90% | 99.90% | 99.90% | 99.90% |
| | $\varepsilon = 0.10$ | 5 iter. (UQ) ↓ | 99.40% | 0.00% | **11.25%** | 92.10% | 48.30% | 65.65% | 43.20% | 69.25% |
| | | 10 iter. (UQ) ↓ | 99.90% | 0.00% | **77.50%** | 98.90% | 98.70% | 98.00% | 96.65% | 98.80% |
| | | 50 iter. (UQ) ↓ | 99.90% | 0.00% | 99.90% | 99.90% | 99.90% | 99.90% | 99.90% | 99.90% |
| | | Breakaway risk ↓ | 0.120% | 0.000% | 0.190% | 0.414% | **0.039%** | 0.146% | 0.211% | 0.254% |
| | | Nearest neighbor acc. ↑ | 0.00% | 95.00% | 0.00% | **0.10%** | **0.10%** | 0.00% | 0.00% | 0.00% |
| | | Overlap risk ↓ | 92.48% | 0.00% | 39.65% | **34.30%** | 45.61% | 40.72% | 93.46% | 91.80% |
| | | Med. adversarial margin ↑ | -18.57% | 84.86% | **4.08%** | 2.72% | 1.23% | 1.60% | -27.18% | -16.50% |
| | | Avg. Certified Radius ↑ | 0.11 | 0.13 | 0.14 | 0.02 | **0.24** | 0.21 | 0.14 | 0.22 |

As customary, we measure the quality of the representations with the top-1 and top-5 accuracy of a linear probe. We also report the accuracy on samples without high-frequency components, as models might be overly reliant on the high-frequency features in the data (Wang et al., 2020). Additionally, we assess the certified robustness via randomized smoothing (Cohen et al., 2019) and report the resulting Average Certified Radius (Zhai et al., 2020).

In line with the impersonation threat model, we also evaluate to what extent attacking a representation encoder can fool a private downstream classifier. Pairs of cats and dogs from the Assira dataset (Elson et al., 2007) are attacked with targeted U-PGD so that the representation of one is close to the other. We report the percentage of impersonations that successfully fool the linear probe.

## 7 RESULTS

In this section, we present the results of the experiments on ResNet50 and the ResNet50-based unsupervised encoders. Further results can be found in App. B.

**Supervised adversarial training performs well on all unsupervised measures.** We validate our unsupervised robustness measures by comparing the robustness at the penultimate layers of ResNet50 and a supervised adversarially trained version of it. The first two columns of Tab. 2 show that adversarially trained ResNet50 scores significantly better on all unsupervised measures. This demonstrates that our measures successfully detect models which we know to be robust in a supervised setting.

**There is no "most robust" standard model.** Amongst the standard unsupervised models, none dominates on all unsupervised robustness measures (see Tab. 2). AMDIM is least susceptible to targeted U-PGD attacks but has the worst untargeted U-PGD, breakaway risk and nearest neighbor accuracy. PixPro significantly outperforms the other models on untargeted attacks. AMDIM and PixPro also have the lowest overlap risk and largest median adversarial margin. At the same time, the model with the lowest breakaway risk and the highest average certified radius is SimCLR. While either AMDIM or PixPro scores the best at most measures, they both have significantly higher

breakaway risk and lower average certified radius than SimCLR. Therefore, no model is a clear choice for the "most robust model".

**Unsupervised robustness measures reveal significant differences among standard models.** The gap between the best and worst performing unsupervised models for the six measures based on targeted U-PGD attacks is between 19% and 27%. The gap reaches almost 81% for the untargeted case (PixPro vs AMDIM, 5 it.), demonstrating that standard models on both extremes do exist. AMDIM has 10.5 times higher breakaway risk than SimCLR while at the same time 2.7 times lower overlap risk than MOCOv2. Observing values on both extremes of all unsupervised robustness metrics testifies to them being useful for differentiating between the different models. Additionally, AMDIM having the highest breakaway risk and lowest overlap risk indicates that unsupervised robustness is a multifaceted problem and that models should be evaluated against an array of measures.

**Unsupervised adversarial training boosts robustness across all measures.** Across every single unsupervised measure, the worst adversarially trained model performs better than the best standard model (Tabs. 2 and 3). Comparing the adversarially trained models with MOCOv2, we observe a significant improvement across the board (Tab. 3). They are also more certifiably robust (Fig. 4). However, the added robustness comes at the price of reduced accuracy (7% to 10%, Tab. 1), as is typical for adversarial training (Zhang et al., 2019; Tsipras et al., 2019). This gap can likely be reduced by fine-tuning the trade-off between the adversarial and standard objectives and by having separate batch normalization parameters for standard and adversarial samples (Kim et al., 2020; Ho & Vasconcelos, 2020). Adversarial training also reduces the impersonation

Table 3: Robustness of MOCOv2 and its adversarially fine-tuned versions measured on PASS and ImageNet.

| | | MOCOv2 | TAR | UNTAR | $\overline{\mathcal{L}}$–PGD |
|---|---|---|---|---|---|
| Targeted U-PGD | $\varepsilon = 0.05$ 5 iter. (RQ) ↑ | 65.64% | 92.97% | 94.00% | **95.41%** |
| | 10 iter. (RQ) ↑ | 52.08% | 87.38% | 88.86% | **91.57%** |
| | 50 iter. (RQ) ↑ | 20.51% | 62.79% | 64.83% | **71.65%** |
| | $\varepsilon = 0.10$ 5 iter. (RQ) ↑ | 68.68% | 93.01% | 93.92% | **95.24%** |
| | 10 iter. (RQ) ↑ | 55.02% | 87.23% | 88.59% | **91.11%** |
| | 50 iter. (RQ) ↑ | 21.88% | 59.67% | 61.21% | **68.19%** |
| Untar. U-PGD | $\varepsilon = 0.05$ 5 iter. (UQ) ↓ | 65.10% | **0.00%** | **0.00%** | **0.00%** |
| | 10 iter. (UQ) ↓ | 99.00% | **0.00%** | **0.00%** | **0.00%** |
| | 50 iter. (UQ) ↓ | 99.90% | 75.35% | 30.35% | **3.40%** |
| | $\varepsilon = 0.10$ 5 iter. (UQ) ↓ | 69.25% | **0.00%** | **0.00%** | **0.00%** |
| | 10 iter. (UQ) ↓ | 98.80% | 0.01% | **0.00%** | **0.00%** |
| | 50 iter. (UQ) ↓ | 99.90% | 91.65% | 65.30% | **18.40%** |
| Breakaway risk ↓ | | 0.254% | 0.049% | **0.001%** | **0.000%** |
| Nearest neighbor acc. ↑ | | 0.00% | 17.00% | 35.60% | **64.40%** |
| Overlap risk ↓ | | 91.80% | **0.00%** | **0.00%** | **0.00%** |
| Med. adversarial margin ↑ | | -16.50% | 61.54% | 67.50% | **76.00%** |
| Avg. Certified Radius ↑ | | 0.22 | 0.30 | 0.30 | **0.31** |

rate of a downstream classifier at 5 iterations by a half relative to MOCOv2 (Tab. 4). For 50 iterations, the rate is similar to MOCOv2 but the attacked images of the adversarially fine-tuned models have more semantically meaningful distortions, which can be detected by a human auditor (see App. D for examples). These results are for only 10 iterations of fine-tuning of a standard encoder. Further impersonation rate reduction can likely be achieved with adversarial training for all 200 epochs.

**Unsupervised adversarial training results in certifiably more robust classifiers.** Fig. 3 shows how the randomized smoothened linear probes of the adversarially trained models uniformly outperform MO-COv2. The difference is especially evident for large radii: 3 times higher certified accuracy when considering perturbations with radius of 0.935. These results demonstrate that unsupervised adversarial training boosts the downstream certified accuracy.

Table 4: Impersonation attack success rate on Assira of MOCOv2 and its label-free adversarially fine-tuned versions for different attack iterations.

| Iterations | MOCOv2 | TAR | UNTAR | $\overline{\mathcal{L}}$–PGD |
|---|---|---|---|---|
| 3 iter. | 34.20% | 18.91% | **15.67%** | **15.20%** |
| 10 iter. | 62.78% | 60.15% | 56.27% | **54.59%** |
| 50 iter. | 76.43% | 76.22% | **71.79%** | **71.89%** |

**Adversarially trained models have better consistency between standard and low-pass accuracy.** The difference between standard and low-pass accuracy for the adversarially trained models is between 1.7% and 2.1%, compared to 2.2% to 9.7% for the standard models (Tab. 1). This could be in part due to the lower accuracy of the adversarially trained models. However, compared with PixPro, AMDIM and SimSiam, which have similar accuracy but larger gaps, indicate that the lower accuracy cannot fully explain the lower gap. Therefore, this suggests that unsupervised adversarial training can help with learning the robust low-frequency features and rejecting high-frequency non-semantic ones.

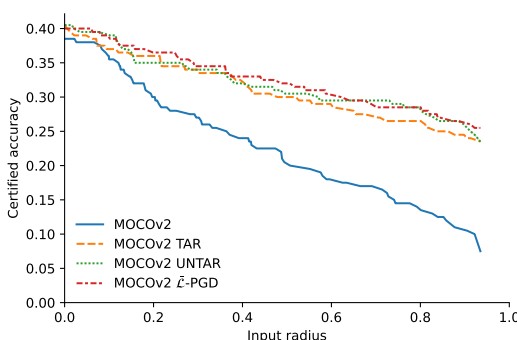 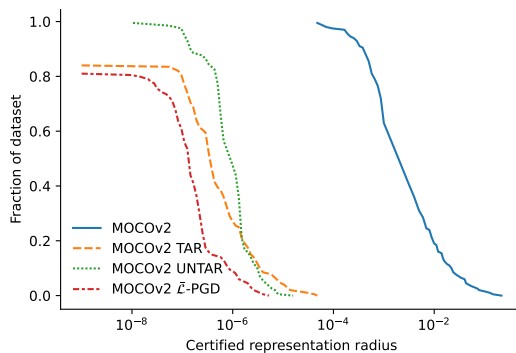

Figure 3: Certified accuracy of randomized smoothed MOCOv2 and its adversarially trained variants on ImageNet. The adversarially trained models are uniformly more robust.

Figure 4: Certified robustness of MOCOv2 on PASS using center smoothing. The certified representation radius is represented as percentile of the distribution of clean representation distances.

$\bar{\mathcal{L}}$-PGD is the overall most robust model, albeit with lower accuracy. $\bar{\mathcal{L}}$-PGD dominates across all unsupervised robustness measures. These results support the findings of prior work on unsupervised adversarial training using batch loss optimization (Ho & Vasconcelos, 2020; Fan et al., 2021; Jiang et al., 2020). However, $\bar{\mathcal{L}}$-PGD also has the lowest supervised accuracy of the three models (Tab. 1), as expected due to the accuracy-robustness trade-off. Still, the differences between the three models are small, and hence all three adversarial training methods can improve the robustness of unsupervised representation learning models.

# 8 DISCUSSION, LIMITATIONS AND CONCLUSION

Unsupervised task-independent adversarial training with simple extensions to classic adversarial attacks can improve the robustness of encoders used for multiple downstream tasks, especially when released publicly. That is why we will release the adversarially fine-tuned versions of MOCOv2 and MOCOv3, which can be used as more robust drop-in replacements for applications built on top of these two models. We showed how to assess the robustness of such encoders without resorting to labeled datasets or proxy tasks. However, there is no single "unsupervised robustness measure": models can have drastically different performance across the different metrics. Still, unsupervised robustness is a stronger requirement than classification robustness as it requires not only the output but also an intermediate state of the model to not be sensitive to small perturbations. Hence, we recommend unsupervised assessment and adversarial training to also be applied to supervised tasks.

We do not compare with the prior methods in Sec. 2 as different base models, datasets and objective functions hinder a fair comparison. Moreover, the methods we propose generalize the previous works, hence this paper strengthens their conclusions rather than claiming improvement over them.

This work is not an exhaustive exploration of unsupervised attacks, robustness measures and defences. We adversarially fine-tuned only three models: two based on ResNet50 and one transformer-based (in App. B); assessing how these techniques work on other architectures is further required. There are many other areas warranting further investigation, such as non-gradient based attacks, measures which better predict the robustness of downstream tasks, certified defences, as well as studying the accuracy-robustness trade-off for representation learning. Still, we believe that robustness evaluation of representation learning models is necessary for a comprehensive assessment of their performance and robustness. This is especially important for encoders used for applications susceptible to impersonation attacks. Therefore, we recommend reporting unsupervised robustness measures together with standard and low-pass linear probe accuracy when proposing new unsupervised and supervised learning models. We hope that this paper illustrates the breadth of opportunities for robustness evaluation in representation space and inspires further work on it.

ETHICS STATEMENT

This work discusses adversarial vulnerabilities in unsupervised models and therefore exposes potential attack vectors for malicious actors. However, it also proposes defence strategies in the form of adversarial training which can alleviate the problem, as well as measures to assess how vulnerable representation learning models are. Therefore, we believe that it would empower the developers of safety-, reliability- and fairness-critical systems to develop safer models. Moreover, we hope that this work inspires further research into unsupervised robustness, which can contribute to more robust and secure machine learning systems.

REPRODUCIBILITY STATEMENT

The experiments in this paper were implemented using open-source software packages (Harris et al., 2020; Virtanen et al., 2020; McKinney, 2010; Paszke et al., 2019), as well as the publicly available MOCOv2 and MOCOv3 models (He et al., 2020; Chen et al., 2020d; 2021). We provide the code used for the adversarial training, as well as all the robustness evaluation implementations, together with documentation on their use. We also release the weights of the models and linear probes. The code reproducing all the experiments in this paper is provided as well. The details are available here.

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

# A    EXAMPLES OF LOSS-BASED ATTACKS

In this appendix we demonstrate how various supervised and unsupervised attacks can be represented as loss-based attacks. This generalizes the unifying view presented by Madry et al. (2018) by incorporating also unsupervised attacks. We also illustrate the resulting structure of which attacks are generalizations of other attacks in Fig. 5. In the following, $\pi(x)$ is the true class of the instance $x$, $\bar{\pi}(x)$ is any other class, $\sigma(x)$ is a function that returns a sample from $\mathcal{D}$ such that $\sigma(x) \neq x$, and $\kappa(x)$ provides a different view of $x$, e.g. a different augmentation.

## A.1    EXAMPLES OF INSTANCE-WISE LOSS-BASED ATTACKS ($\mathcal{L}$-FGSM, $\mathcal{L}$-PGD, $\mathcal{L}$-BIM)

- The supervised FGSM and PGD attacks can be considered as special cases of the unsupervised $\mathcal{L}$-FGSM and $\mathcal{L}$-PGD, where $f$ is a classifier rather than an encoder and we use a classification loss:
  - Untargeted FGSM attack: $\mathcal{L}$-FGSM with $\mathcal{L}(f, x) = \mathcal{L}_{\mathrm{cl}}(f(x), \pi(x))$,
  - Targeted FGSM attack: $\mathcal{L}$-FGSM with $\mathcal{L}(f, x) = -\mathcal{L}_{\mathrm{cl}}(f(x), \bar{\pi}(x))$,
  - Untargeted PGD attack: $\mathcal{L}$-PGD with $\mathcal{L}(f, x, \hat{x}_u) = \mathcal{L}_{\mathrm{cl}}(f(\hat{x}_u), \pi(x))$,
  - Targeted PGD attack: $\mathcal{L}$-PGD with $\mathcal{L}(f, x, \hat{x}_u) = -\mathcal{L}_{\mathrm{cl}}(f(\hat{x}_u), \bar{\pi}(x))$.

- The U-LGSM and U-PGD attacks can be represented as $\mathcal{L}$-FGSM and $\mathcal{L}$-PGD attacks where the loss is the divergence $d$:
  - Untargeted U-FGSM attack: $\mathcal{L}$-FGSM with $\mathcal{L}(f, x) = d(f(x), f(x + \eta))$,
  - Targeted U-FGSM attack: $\mathcal{L}$-FGSM with $\mathcal{L}(f, x) = -d(f(x), f(\sigma(x)))$,
  - Untargeted U-PGD attack: $\mathcal{L}$-PGD with $\mathcal{L}(f, x, \hat{x}_u) = d(f(\hat{x}_u), f(x))$,
  - Targeted U-PGD attack: $\mathcal{L}$-PGD with $\mathcal{L}(f, x, \hat{x}_u) = -d(f(\hat{x}_u), f(\sigma(x)))$.

- Untargeted U-PGD with the Kullback-Leibler divergence corresponds to the adversarial example generation process of UAT-OT (Alayrac et al., 2019) which is based on the Virtual Adversarial Training method for semi-supervised adversarial learning (Miyato et al., 2019). Untargeted U-PGD with the Kullback-Leibler also corresponds to the robustness regularizer proposed by Nguyen et al. (2022).

- Cemgil et al. (2020) propose an unsupervised attack for Variational Auto-Encoders (VAEs) (Kingma & Welling, 2014) based on the Wasserstein distance. It can be represented as the untargeted U-PGD attack with the Wasserstein distance, or equivalently, as the $\mathcal{L}$-PGD attack with the loss

$$\mathcal{L}(f, x, \hat{x}_u) = \mathcal{W}\left(\mathcal{N}([f(x)]_{\boldsymbol{\mu}}, \boldsymbol{I}[f(x)]_{\boldsymbol{\sigma}}), \mathcal{N}[(f(\hat{x}_u)]_{\boldsymbol{\mu}}, \boldsymbol{I}[f(\hat{x}_u)]_{\boldsymbol{\sigma}})\right),$$

  where $\mathcal{W}$ is the Wasserstein distance, $\mathcal{N}$ is the normal distribution, $\boldsymbol{I}$ is the identity matrix and the subscripts $\boldsymbol{\mu}$ and $\boldsymbol{\sigma}$ designate the respective outputs of the VAE encoder $f$.

- The BYORL method by Gowal et al. (2020) is equivalent to the untargeted U-PGD attack when the divergence $d$ is chosen to be the cosine similarity.

- The instance-wise unsupervised adversarial attack proposed by Kim et al. (2020) is equivalent to $\mathcal{L}$-PGD with the contrastive loss

$$\mathcal{L}(f, x, \hat{x}_u) = -\log \frac{\exp\left(f(\hat{x}_u)^{\top} f(\kappa(x))/T\right)}{\exp\left(f(\hat{x}_u)^{\top} f(\kappa(x))/T\right) + \exp\left(f(\hat{x}_u)^{\top} f(\sigma(x))/T\right)},$$

  where $T$ is a temperature parameter. This loss encourages that the cosine similarity between the adversarial example and another view of the same sample is small relative to the cosine similarity between the adversarial example and another sample from $\mathcal{D}$.

- The concurrent work by Kim et al. (2022) proposing targeted adversarial self-supervised learning can also be considered to be an instance of the targeted $\mathcal{L}$-PGD attack.

- Using the NT-Xent loss with $\mathcal{L}$-PGD results in the attack used for the Adversarial-to-Standard adversarial contrastive learning proposed by Jiang et al. (2020).

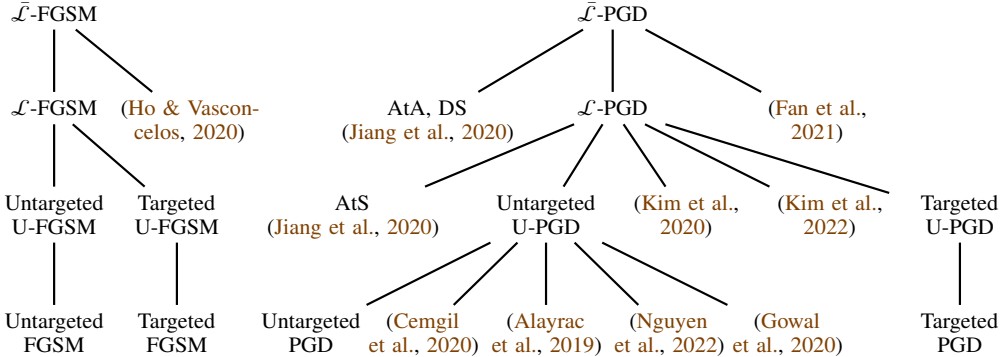

Figure 5: The hierarchy of supervised and unsupervised attacks.

## A.2 EXAMPLES OF BATCH-WISE LOSS-BASED ATTACKS ($\bar{\mathcal{L}}$-FGSM, $\bar{\mathcal{L}}$-PGD, $\bar{\mathcal{L}}$-BIM)

- Any $\mathcal{L}$-FGSM attack with loss $\mathcal{L}$ is trivially an $\bar{\mathcal{L}}$-FGSM attack by taking $N = 1$ and considering the loss function $\bar{\mathcal{L}} = \mathcal{L}$.

- Similarly, any $\mathcal{L}$-PGD attack is trivially an $\bar{\mathcal{L}}$-PGD attack.

- The adversarial attack proposed by Ho & Vasconcelos (2020) is equivalent to $\bar{\mathcal{L}}$-FGSM with the contrastive loss

$$\bar{\mathcal{L}}(f, X) = \sum_{i=1}^{N} -\log \frac{\exp(f(x_i)^\top f(\kappa(x_i))/T)}{\sum_{j=1}^{N} \exp(f(x_j)^\top f(\kappa(x_i))/T)}.$$

Here adversarial examples are selected to *jointly* maximize the contrastive loss.

- The adversarial training of AdvCL generates adversarial attacks by maximizing a multi-view contrastive loss computed over the adversarial example, two views of $x$ and its high-frequency component $\text{HighPass}(x)$ (Fan et al., 2021). It corresponds to $\bar{\mathcal{L}}$-PGD with the loss

$$\mathcal{L}(f, X, \hat{X}_u) = \frac{1}{N} \sum_{i=1}^{N} \mathcal{L}'\left(\kappa_1(x_i), \kappa_2(x_i), \hat{x}_{i,u}, \text{HighPass}(x_i); f, X\right),$$

$$\mathcal{L}'(z_1, \ldots, z_m; f, X) = -\sum_{i=1}^{m} \sum_{\substack{j=1 \\ j \neq i}}^{m} \log \frac{\exp(\text{sim}(f(z_i), f(z_j))/T)}{\sum_{z_k \in X} \sum_{\kappa \in \{\kappa_1, \kappa_2\}} \exp(\text{sim}(f(z_i), f(\kappa(z_k)))/T)},$$

with $\text{sim}(\cdot, \cdot)$ being the cosine similarity.

- Using the NT-Xent loss and the $\bar{\mathcal{L}}$-PGD attack on a pair of views of $x$ is identical to the Adversarial-to-Adversarial and Dual Stream adversarial contrastive learning methods by Jiang et al. (2020).

## B EXTENDED RESULTS

This appendix presents more comprehensive experimental results in Tabs. 5 to 7 and Figs. 6 to 11.

We consider ResNet50 (penultimate layer) (He et al., 2016), a supervised adversarially trained ResNet50 ($\ell_\infty, \epsilon = 4/255$, penultimate layer) (Salman et al., 2020) and the ResNet50-based self-supervised learning models MOCOv2 (200 epochs) (He et al., 2020; Chen et al., 2020d), MOCO with non-semantic negatives (+Patch, $k$=16384, $\alpha$=3) (Ge et al., 2021), PixPro (400 epochs) (Xie et al., 2021), AMDIM (Medium) (Bachman et al., 2019), SimCLRv2 (depth 50, width 1x, without selective kernels) (Chen et al., 2020c), and SimSiam (100 epochs, batch size 256) (Chen & He, 2021). In addition to the ResNet50-based models, we also present results from two models with transformer architectures (Vaswani et al., 2017). MAE uses masked autoencoders (He et al., 2022) and we use its

Table 5: Extended results for ResNet50-based self-supervised models and unsupervised adversarially fine-tuned MOCOv2. Arrows show if larger or smaller values are better. UQ and RQ respectively designate values reported in universal and relative quantiles.

| | | Standard ResNet-based unsupervised models | | | | | | Unsup. adv. fine-tuned MOCOv2 | | |
|---|---|---|---|---|---|---|---|---|---|---|
| | | PixPro | AMDIM | SimCLR | SimSiam | Nonsem | MOCOv2 | TAR | UNTAR | $\bar{\mathcal{L}}$–PGD |
| Top-1 accuracy ↑ | | 58.0% | 62.3% | 67.2% | 62.7% | 49.7% | 67.4% | 60.2% | 59.9% | 57.2% |
| Top-5 accuracy ↑ | | 80.5% | 82.2% | 86.6% | 85.2% | 73.5% | 87.7% | 82.4% | 82.2% | 80.0% |
| Lowpass Top-1 accuracy ↑ | | 50.0% | 52.7% | 59.7% | 56.3% | 47.3% | 62.2% | 58.6% | 58.2% | 55.4% |
| Lowpass Top-5 accuracy ↑ | | 73.3% | 73.9% | 81.2% | 80.2% | 71.2% | 84.0% | 81.2% | 80.7% | 78.4% |
| **Targeted U-PGD attack — ε=0.05 — $\ell_2$ distance** | 5 iter. (RQ) ↑ | 80.5% | 80.8% | 75.0% | 73.3% | 61.5% | 65.6% | 93.0% | 94.0% | 95.4% |
| | 10 iter. (RQ) ↑ | 67.9% | 70.3% | 65.8% | 65.9% | 46.5% | 52.1% | 87.4% | 88.9% | 91.6% |
| | 30 iter. (RQ) ↑ | 45.6% | 52.5% | 45.3% | 44.1% | 28.3% | 29.3% | 72.4% | 74.3% | 79.9% |
| | 50 iter. (RQ) ↑ | 37.6% | 44.3% | 36.4% | 28.1% | 22.2% | 20.5% | 62.8% | 64.8% | 71.6% |
| **$\ell_\infty$ distance** | 5 iter. (RQ) ↑ | 85.5% | 62.0% | 89.4% | 76.0% | 62.2% | 65.2% | 93.3% | 94.3% | 95.6% |
| | 10 iter. (RQ) ↑ | 69.4% | 46.8% | 76.7% | 66.9% | 46.0% | 52.3% | 87.8% | 89.3% | 91.9% |
| | 30 iter. (RQ) ↑ | 39.9% | 29.4% | 43.4% | 45.7% | 27.3% | 29.2% | 72.5% | 74.9% | 80.2% |
| | 50 iter. (RQ) ↑ | 30.7% | 24.2% | 32.7% | 28.4% | 21.1% | 20.5% | 62.9% | 65.2% | 72.0% |
| **Cosine similarity** | 5 iter. ↓ | 0.89 | 0.49 | 0.40 | 0.09 | 0.98 | 0.57 | 0.29 | 0.29 | 0.32 |
| | 10 iter. ↓ | 0.92 | 0.59 | 0.57 | 0.18 | 0.99 | 0.75 | 0.34 | 0.34 | 0.36 |
| | 30 iter. ↓ | 0.97 | 0.78 | 0.83 | 0.74 | 1.00 | 0.93 | 0.51 | 0.50 | 0.47 |
| | 50 iter. ↓ | 0.98 | 0.84 | 0.90 | 0.91 | 1.00 | 0.97 | 0.64 | 0.62 | 0.56 |
| **ε=0.10 — $\ell_2$ distance** | 5 iter. (RQ) ↑ | 83.1% | 85.6% | 78.0% | 78.1% | 64.3% | 68.7% | 93.0% | 93.9% | 95.2% |
| | 10 iter. (RQ) ↑ | 71.2% | 75.9% | 69.0% | 70.4% | 48.6% | 55.0% | 87.2% | 88.6% | 91.1% |
| | 30 iter. (RQ) ↑ | 47.2% | 56.4% | 48.6% | 48.0% | 29.0% | 31.4% | 70.9% | 72.5% | 77.9% |
| | 50 iter. (RQ) ↑ | 38.1% | 47.3% | 38.6% | 30.3% | 22.3% | 21.9% | 59.7% | 61.2% | 68.2% |
| **$\ell_\infty$ distance** | 5 iter. (RQ) ↑ | 89.0% | 68.0% | 90.9% | 78.6% | 65.8% | 68.8% | 93.1% | 94.1% | 95.3% |
| | 10 iter. (RQ) ↑ | 74.2% | 54.0% | 80.5% | 69.4% | 48.3% | 55.4% | 87.3% | 88.9% | 91.2% |
| | 30 iter. (RQ) ↑ | 42.1% | 30.7% | 48.6% | 48.0% | 28.0% | 31.4% | 70.9% | 72.8% | 78.2% |
| | 50 iter. (RQ) ↑ | 31.2% | 24.0% | 35.3% | 29.8% | 21.2% | 22.1% | 60.3% | 62.1% | 68.2% |
| **Cosine similarity** | 5 iter. ↓ | 0.88 | 0.41 | 0.36 | 0.09 | 0.98 | 0.53 | 0.29 | 0.30 | 0.32 |
| | 10 iter. ↓ | 0.92 | 0.50 | 0.52 | 0.17 | 0.99 | 0.72 | 0.34 | 0.35 | 0.36 |
| | 30 iter. ↓ | 0.97 | 0.71 | 0.80 | 0.68 | 1.00 | 0.92 | 0.54 | 0.53 | 0.49 |
| | 50 iter. ↓ | 0.98 | 0.80 | 0.88 | 0.90 | 1.00 | 0.96 | 0.69 | 0.67 | 0.61 |
| **Untargeted U-PGD attack — ε=0.05 — $\ell_2$ distance** | 5 iter. (UQ) ↓ | 14.4% | 81.3% | 43.7% | 72.6% | 46.8% | 65.1% | 0.0% | 0.0% | 0.0% |
| | 10 iter. (UQ) ↓ | 85.6% | 98.5% | 99.4% | 98.6% | 98.1% | 99.0% | 0.0% | 0.0% | 0.0% |
| | 30 iter. (UQ) ↓ | 99.9% | 99.9% | 99.9% | 99.9% | 99.9% | 99.9% | 9.8% | 0.7% | 0.0% |
| | 50 iter. (UQ) ↓ | 99.9% | 99.9% | 99.9% | 99.9% | 99.9% | 99.9% | 75.4% | 30.4% | 3.4% |
| **$\ell_\infty$ distance** | 5 iter. (UQ) ↓ | 25.0% | 76.4% | 64.2% | 43.8% | 51.7% | 62.3% | 0.0% | 0.0% | 0.0% |
| | 10 iter. (UQ) ↓ | 72.7% | 95.7% | 89.9% | 96.8% | 97.7% | 98.7% | 0.0% | 0.0% | 0.0% |
| | 30 iter. (UQ) ↓ | 97.4% | 99.9% | 99.8% | 99.9% | 99.9% | 99.9% | 15.1% | 2.0% | 0.1% |
| | 50 iter. (UQ) ↓ | 98.7% | 99.9% | 99.9% | 99.9% | 99.9% | 99.9% | 73.0% | 39.0% | 5.7% |
| **Cosine similarity** | 5 iter. ↑ | 0.88 | 0.44 | 0.63 | 0.30 | 0.97 | 0.49 | 0.98 | 0.98 | 0.99 |
| | 10 iter. ↑ | 0.78 | 0.32 | 0.52 | 0.16 | 0.91 | 0.31 | 0.93 | 0.95 | 0.97 |
| | 30 iter. ↑ | 0.62 | 0.11 | 0.40 | 0.08 | 0.78 | 0.18 | 0.69 | 0.75 | 0.85 |
| | 50 iter. ↑ | 0.56 | 0.04 | 0.36 | 0.06 | 0.73 | 0.15 | 0.52 | 0.59 | 0.72 |
| **ε=0.10 — $\ell_2$ distance** | 5 iter. (UQ) ↓ | 11.3% | 92.1% | 48.3% | 65.7% | 43.2% | 69.3% | 0.0% | 0.0% | 0.0% |
| | 10 iter. (UQ) ↓ | 77.5% | 98.9% | 98.7% | 98.0% | 96.6% | 98.8% | 0.0% | 0.0% | 0.0% |
| | 30 iter. (UQ) ↓ | 99.9% | 99.9% | 99.9% | 99.9% | 99.9% | 99.9% | 30.6% | 6.1% | 0.4% |
| | 50 iter. (UQ) ↓ | 99.9% | 99.9% | 99.9% | 99.9% | 99.9% | 99.9% | 91.7% | 65.3% | 18.4% |
| **$\ell_\infty$ distance** | 5 iter. (UQ) ↓ | 24.6% | 87.1% | 67.1% | 43.8% | 49.1% | 63.6% | 0.0% | 0.0% | 0.0% |
| | 10 iter. (UQ) ↓ | 69.6% | 97.3% | 89.5% | 95.6% | 96.0% | 98.0% | 0.0% | 0.0% | 0.0% |
| | 30 iter. (UQ) ↓ | 97.4% | 99.9% | 99.8% | 99.9% | 99.9% | 99.9% | 37.7% | 11.2% | 1.0% |
| | 50 iter. (UQ) ↓ | 98.9% | 99.9% | 99.9% | 99.9% | 99.9% | 99.9% | 90.2% | 66.6% | 23.7% |
| **Cosine similarity** | 5 iter. ↑ | 0.89 | 0.37 | 0.54 | 0.21 | 0.96 | 0.41 | 0.94 | 0.95 | 0.98 |
| | 10 iter. ↑ | 0.79 | 0.27 | 0.45 | 0.08 | 0.91 | 0.26 | 0.88 | 0.90 | 0.95 |
| | 30 iter. ↑ | 0.62 | 0.08 | 0.35 | 0.03 | 0.77 | 0.15 | 0.57 | 0.64 | 0.78 |
| | 50 iter. ↑ | 0.55 | 0.00 | 0.32 | 0.02 | 0.71 | 0.13 | 0.39 | 0.45 | 0.60 |
| Breakaway risk ↓ | | 0.190% | 0.414% | 0.039% | 0.146% | 0.211% | 0.254% | 0.049% | 0.001% | 0.000% |
| Nearest neighbor accuracy ↑ | | 0.0% | 0.1% | 0.1% | 0.0% | 0.0% | 0.0% | 17.0% | 35.6% | 64.4% |
| Overlap risk ↓ | | 39.6% | 34.3% | 45.6% | 40.7% | 93.5% | 91.8% | 0.0% | 0.0% | 0.0% |
| Median adversarial margin ↑ | | 4.1% | 2.7% | 1.2% | 1.6% | -27.2% | -16.5% | 61.5% | 67.5% | 76.0% |
| Average Certified Radius ↑ | | 0.14 | 0.02 | 0.24 | 0.21 | 0.14 | 0.22 | 0.30 | 0.30 | 0.31 |
| **Impersonation rate** | 3 iter. ↓ | 43.0% | 23.2% | 40.0% | 21.0% | 48.7% | 34.2% | 18.9% | 15.7% | 15.2% |
| | 10 iter. ↓ | 71.3% | 71.9% | 74.5% | 52.4% | 75.5% | 62.8% | 60.1% | 56.3% | 54.6% |
| | 30 iter. ↓ | 83.3% | 88.9% | 85.6% | 76.3% | 83.9% | 74.4% | 75.1% | 69.5% | 70.0% |
| | 50 iter. ↓ | 86.7% | 91.0% | 88.6% | 80.6% | 85.6% | 76.4% | 76.2% | 71.8% | 71.9% |

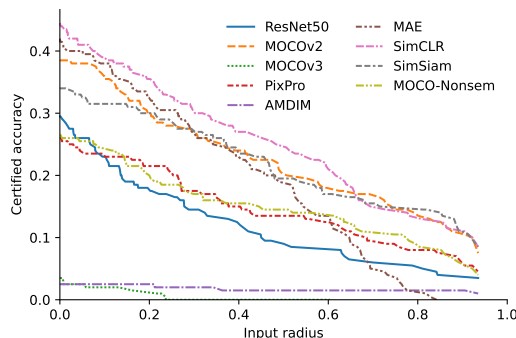 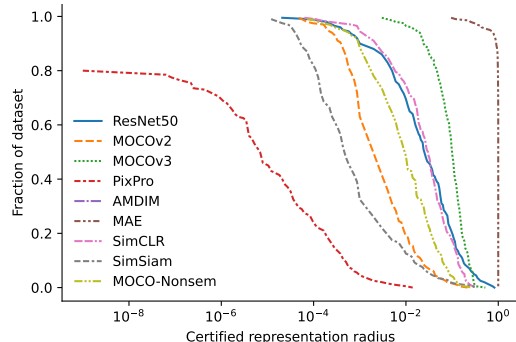

Figure 6: Certified accuracy of the standard models on ImageNet.

Figure 7: Certified robustness of the standard models on PASS using center smoothing. The distribution of certified radii in $\mathcal{R}$ is reported as percentile of the clean representation distances. Smaller values indicate higher unsupervised robustness.

ViT-Large variant. MOCOv3 is a modification of MOCOv2 to work with a transformer backbone (ViT-Small) (Chen et al., 2021).

We also apply the targeted and untargeted unsupervised adversarial fine-tuning techniques from the main text to the ResNet50 encoder and the targeted, untargeted and loss-based unsupervised adversarial fine-tuning to MOCOv3 to compare how our findings from Sec. 7 transfer to supervised learning encoders and transformers. We use the exact same attack types and parameter values as for MOCOv2. The implementation details are in App. C.

For all models, we present top-1 and top-5 accuracy for both the standard and the lowpass settings in Tabs. 5 to 7. We report the results for the $\ell_2$- and $\ell_\infty$-induced divergences in representation space, at iterations 5, 10, 30 and 50 for U-PGD attacks with both $\epsilon = 0.05$ and $\epsilon = 0.10$. The attacks are performed with $d(x, x') = \|x - x'\|_2$ and $\alpha = 0.001$. These are reported as universal quantiles for the untargeted attacks and relative quantiles for the targeted attacks. We also report the breakaway and overlap risks, as well as the nearest neighbor accuracy, median adversarial margin, average certified radius and impersonation rates as in the main text. Figs. 6 to 11 show the certified robustness and accuracy of the models which were omitted from the main text.

Some of the models, including MOCOv2, are trained with a contrastive objective based on the cosine similarity. Therefore, using the Euclidean distances as the divergence in representation space could be considered an unfair comparison as the adversarially trained models are optimized for it while the standard models are optimized for the cosine similarity. Therefore, for targeted attacks, we also report the median cosine similarity between the representations of the adversarial examples and the representations of the target samples. Higher values mean that the attack is more successful and that the model is less robust. For the untargeted attacks, we report the median cosine similarity between the representations of the adversarial examples and the representations of the original samples. Hence, higher values mean that the attack is less successful and the model is more robust. The results in Tabs. 5 to 7 show that adversarial training with the $\ell_2$-induced divergence leads also to improvements when measuring the cosine similarity: the divergence that the standard models are trained for but the adversarial ones are not. This evidence supports our claim that the improvements we see from unsupervised adversarial fine-tuning are not due to our choice of divergence.

## B.1 Supervised backbone models based on ResNet50

In Tab. 6 and Figs. 8 and 9 we compare the penultimate layer of ResNet50, the penultimate layer of ResNet50 trained with supervised adversarial training, as well as fine-tuned ResNet50 (same as the first model) using our targeted and untargeted U-PGD attacks. We did not consider $\mathcal{L}$-PGD as ResNet does not have a corresponding loss in representation space due to its supervised nature. The technical details are in App. C.3.

Table 6: Extended results for ResNet50, supervised adversarially trained ResNet50, and unsupervised adversarially fine-tuned ResNet50. Arrows show if larger or smaller values are better. UQ and RQ respectively designate values reported in universal and relative quantiles.

| | ResNet50 | Sup. Robust ResNet50 | Unsup. fine-tuned ResNet50 TAR | Unsup. fine-tuned ResNet50 UNTAR |
|---|---|---|---|---|
| Top-1 accuracy ↑ | 74.1% | 62.0% | 65.2% | 54.2% |
| Top-5 accuracy ↑ | 91.0% | 82.1% | 84.1% | 76.2% |
| Lowpass Top-1 accuracy ↑ | 67.7% | 59.2% | 58.0% | 49.9% |
| Lowpass Top-5 accuracy ↑ | 86.4% | 79.8% | 78.3% | 72.3% |
| **Targeted U-PGD attack, ε = 0.05** | | | | |
| $\ell_2$ distance — 5 iter. (RQ) ↑ | 66.4% | 98.1% | 69.5% | 88.4% |
| $\ell_2$ distance — 10 iter. (RQ) ↑ | 52.7% | 96.5% | 57.1% | 80.6% |
| $\ell_2$ distance — 30 iter. (RQ) ↑ | 33.5% | 91.0% | 37.4% | 62.2% |
| $\ell_2$ distance — 50 iter. (RQ) ↑ | 27.4% | 86.9% | 30.8% | 52.0% |
| $\ell_\infty$ distance — 5 iter. (RQ) ↑ | 70.3% | 98.6% | 72.8% | 92.5% |
| $\ell_\infty$ distance — 10 iter. (RQ) ↑ | 52.3% | 97.3% | 57.8% | 85.6% |
| $\ell_\infty$ distance — 30 iter. (RQ) ↑ | 31.3% | 92.5% | 35.8% | 65.0% |
| $\ell_\infty$ distance — 50 iter. (RQ) ↑ | 24.5% | 88.8% | 28.7% | 53.1% |
| Cosine similarity — 5 iter. ↓ | 0.77 | 0.47 | 0.76 | 0.97 |
| Cosine similarity — 10 iter. ↓ | 0.86 | 0.48 | 0.85 | 0.97 |
| Cosine similarity — 30 iter. ↓ | 0.95 | 0.51 | 0.94 | 0.98 |
| Cosine similarity — 50 iter. ↓ | 0.97 | 0.54 | 0.96 | 0.99 |
| **Targeted U-PGD attack, ε = 0.10** | | | | |
| $\ell_2$ distance — 5 iter. (RQ) ↑ | 69.3% | 97.9% | 81.3% | 89.2% |
| $\ell_2$ distance — 10 iter. (RQ) ↑ | 55.6% | 96.0% | 70.0% | 81.4% |
| $\ell_2$ distance — 30 iter. (RQ) ↑ | 34.7% | 89.4% | 46.4% | 61.1% |
| $\ell_2$ distance — 50 iter. (RQ) ↑ | 27.9% | 83.9% | 36.9% | 49.3% |
| $\ell_\infty$ distance — 5 iter. (RQ) ↑ | 74.3% | 98.6% | 90.1% | 93.2% |
| $\ell_\infty$ distance — 10 iter. (RQ) ↑ | 58.0% | 97.3% | 78.0% | 86.2% |
| $\ell_\infty$ distance — 30 iter. (RQ) ↑ | 34.5% | 92.3% | 46.7% | 63.7% |
| $\ell_\infty$ distance — 50 iter. (RQ) ↑ | 27.0% | 86.6% | 34.8% | 48.3% |
| Cosine similarity — 5 iter. ↓ | 0.74 | 0.47 | 0.67 | 0.97 |
| Cosine similarity — 10 iter. ↓ | 0.84 | 0.48 | 0.76 | 0.97 |
| Cosine similarity — 30 iter. ↓ | 0.94 | 0.53 | 0.90 | 0.98 |
| Cosine similarity — 50 iter. ↓ | 0.96 | 0.57 | 0.94 | 0.99 |
| **Untargeted U-PGD attack, ε = 0.05** | | | | |
| $\ell_2$ distance — 5 iter. (UQ) ↓ | 98.7% | 0.0% | 93.4% | 0.0% |
| $\ell_2$ distance — 10 iter. (UQ) ↓ | 99.9% | 0.0% | 99.9% | 0.7% |
| $\ell_2$ distance — 30 iter. (UQ) ↓ | 99.9% | 0.0% | 99.9% | 21.4% |
| $\ell_2$ distance — 50 iter. (UQ) ↓ | 99.9% | 0.0% | 99.9% | 47.6% |
| $\ell_\infty$ distance — 5 iter. (UQ) ↓ | 74.1% | 0.0% | 88.8% | 0.8% |
| $\ell_\infty$ distance — 10 iter. (UQ) ↓ | 99.3% | 0.0% | 97.9% | 4.7% |
| $\ell_\infty$ distance — 30 iter. (UQ) ↓ | 99.9% | 0.0% | 99.9% | 29.2% |
| $\ell_\infty$ distance — 50 iter. (UQ) ↓ | 99.9% | 0.0% | 99.9% | 49.9% |
| Cosine similarity — 5 iter. ↑ | 0.72 | 0.99 | 0.69 | 1.00 |
| Cosine similarity — 10 iter. ↑ | 0.64 | 0.99 | 0.63 | 0.99 |
| Cosine similarity — 30 iter. ↑ | 0.56 | 0.95 | 0.56 | 0.98 |
| Cosine similarity — 50 iter. ↑ | 0.52 | 0.91 | 0.53 | 0.96 |
| **Untargeted U-PGD attack, ε = 0.10** | | | | |
| $\ell_2$ distance — 5 iter. (UQ) ↓ | 99.4% | 0.0% | 34.4% | 0.4% |
| $\ell_2$ distance — 10 iter. (UQ) ↓ | 99.9% | 0.0% | 97.6% | 2.5% |
| $\ell_2$ distance — 30 iter. (UQ) ↓ | 99.9% | 0.1% | 99.9% | 34.7% |
| $\ell_2$ distance — 50 iter. (UQ) ↓ | 99.9% | 0.9% | 99.9% | 65.4% |
| $\ell_\infty$ distance — 5 iter. (UQ) ↓ | 78.6% | 0.0% | 84.9% | 1.9% |
| $\ell_\infty$ distance — 10 iter. (UQ) ↓ | 99.0% | 0.0% | 93.3% | 6.9% |
| $\ell_\infty$ distance — 30 iter. (UQ) ↓ | 99.9% | 0.4% | 99.9% | 39.6% |
| $\ell_\infty$ distance — 50 iter. (UQ) ↓ | 99.9% | 4.3% | 99.9% | 64.6% |
| Cosine similarity — 5 iter. ↑ | 0.68 | 0.95 | 0.69 | 0.99 |
| Cosine similarity — 10 iter. ↑ | 0.62 | 0.94 | 0.63 | 0.99 |
| Cosine similarity — 30 iter. ↑ | 0.54 | 0.86 | 0.55 | 0.97 |
| Cosine similarity — 50 iter. ↑ | 0.52 | 0.78 | 0.53 | 0.94 |
| Breakaway risk ↓ | 0.120% | 0.000% | 0.073% | 0.111% |
| Nearest neighbor accuracy ↑ | 0.0% | 95.0% | 0.0% | 2.0% |
| Overlap risk ↓ | 92.5% | 0.0% | 78.5% | 0.9% |
| Median adversarial margin ↑ | -18.6% | 84.9% | -9.8% | 47.6% |
| Average Certified Radius ↑ | 0.11 | 0.13 | 0.22 | 0.16 |
| Impersonation rate — 3 iter. ↓ | 41.5% | 6.4% | 25.3% | 25.3% |
| Impersonation rate — 10 iter. ↓ | 76.5% | 49.9% | 70.2% | 63.6% |
| Impersonation rate — 30 iter. ↓ | 87.1% | 83.7% | 85.3% | 80.7% |
| Impersonation rate — 50 iter. ↓ | 89.9% | 86.9% | 88.3% | 84.6% |

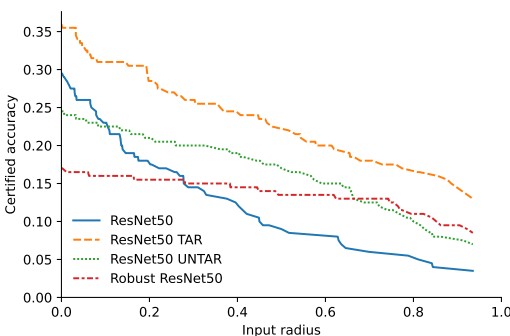

Figure 8: Certified accuracy of Robust ResNet50, ResNet50 and its adv. trained variants on ImageNet computed via randomized smoothing. The adversarially trained models are uniformly certifiably more robust for almost all radius values.

Figure 9: Certified robustness of ResNet50 on PASS using center smoothing. The distribution of certified radii in $\mathcal{R}$ is reported as a percentile of the clean representation distances. Smaller values indicate higher unsupervised robustness.

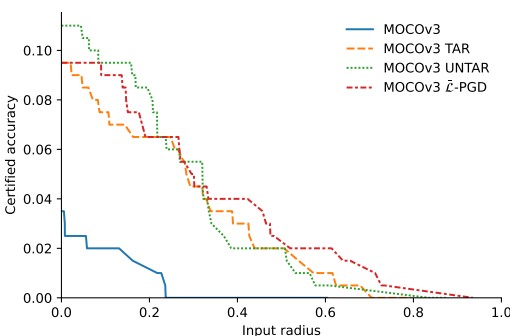

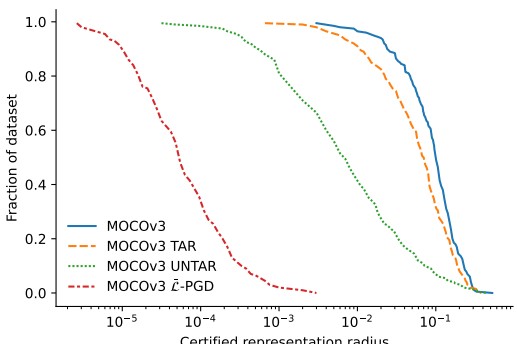

Figure 10: Certified accuracy of MOCOv3 and its adversarially trained variants on ImageNet computed via randomized smoothing. The adversarially trained models are uniformly certifiably more robust for almost all radius values.

Figure 11: Certified robustness of MOCOv3 on PASS using center smoothing. The distribution of certified radii in $\mathcal{R}$ is reported as a percentile of the clean representation distances. Smaller values indicate higher unsupervised robustness.

As mentioned in Sec. 7, the supervised adversarially trained ResNet50 has by far the best robustness scores across all measures. It also significantly surpasses our unsupervised adversarially fine-tuned models. However, this is not enough to conclude that supervised adversarial training works better than unsupervised adversarial training. The supervised adversarially trained ResNet50 has been trained with attacks for 150 epochs, while our unsupervised adversarially trained models were fine-tuned for 10 epochs. Therefore, the gap in robustness could be due to the quantity of adversarial training rather than the quality of the technique.

Both the targeted and untargeted unsupervised adversarially fine-tuned ResNet50 models score better on all robustness measures than the standard ResNet50. The improvements are markedly larger for the untargeted fine-tuned model. Though this comes at the price of lower accuracy and lower certified accuracy (Fig. 8). Furthermore, both models have very similar certified robustness, as shown in Fig. 9. Both the supervised adversarially trained and the unsupervised adversarially fine-tuned with untargeted attacks models have lower certified accuracy than the standard model at lower radii. However, they compensate that with higher certified accuracy for larger radii resulting in overall improvement in the average certified radius (Tab. 6).

## B.2 Transformer backbone models: MAE and MOCOv3

Among the standard models, MAE outperforms the standard models on most measures. MAE has the highest accuracy of all models trained in the unsupervised regime, i.e. excluding the supervised

Table 7: Extended results for transformer-based models and adversarially fine-tuned MOCOv3. Arrows show if larger or smaller values are better. UQ and RQ respectively designate values reported in universal and relative quantiles.

| | | | Transformer models | | Unsup. adv. fine-tuned MOCOv3 | | |
|---|---|---|---|---|---|---|---|
| | | | MAE | MOCOv3 | TAR | UNTAR | $\overline{\mathcal{L}}$–PGD |
| | | Top-1 accuracy ↑ | 71.8% | 68.6% | 67.9% | 66.5% | 65.3% |
| | | Top-5 accuracy ↑ | 88.9% | 87.7% | 87.5% | 86.6% | 85.8% |
| | | Lowpass Top-1 accuracy ↑ | 66.2% | 63.4% | 64.2% | 63.3% | 62.6% |
| | | Lowpass Top-5 accuracy ↑ | 85.1% | 83.8% | 84.7% | 84.0% | 83.4% |
| Targeted U-PGD attack | ε = 0.05 | $\ell_2$ distance — 5 iter. (RQ) ↑ | 80.6% | 72.0% | 86.2% | 87.1% | 89.8% |
| | | $\ell_2$ distance — 10 iter. (RQ) ↑ | 66.3% | 63.2% | 77.3% | 79.3% | 82.3% |
| | | $\ell_2$ distance — 30 iter. (RQ) ↑ | 42.2% | 31.0% | 50.6% | 49.9% | 47.6% |
| | | $\ell_2$ distance — 50 iter. (RQ) ↑ | 33.7% | 20.3% | 31.3% | 30.6% | 26.2% |
| | | $\ell_\infty$ distance — 5 iter. (RQ) ↑ | 72.4% | 72.6% | 87.9% | 89.0% | 90.5% |
| | | $\ell_\infty$ distance — 10 iter. (RQ) ↑ | 54.4% | 62.4% | 78.5% | 81.2% | 82.8% |
| | | $\ell_\infty$ distance — 30 iter. (RQ) ↑ | 27.6% | 30.6% | 52.3% | 51.3% | 48.3% |
| | | $\ell_\infty$ distance — 50 iter. (RQ) ↑ | 20.0% | 19.7% | 32.1% | 30.4% | 26.2% |
| | | Cosine similarity — 5 iter. ↓ | 0.99 | 0.26 | 0.06 | 0.06 | 0.06 |
| | | Cosine similarity — 10 iter. ↓ | 0.99 | 0.44 | 0.14 | 0.14 | 0.15 |
| | | Cosine similarity — 30 iter. ↓ | 1.00 | 0.91 | 0.65 | 0.70 | 0.76 |
| | | Cosine similarity — 50 iter. ↓ | 1.00 | 0.96 | 0.89 | 0.91 | 0.94 |
| | ε = 0.10 | $\ell_2$ distance — 5 iter. (RQ) ↑ | 83.6% | 76.1% | 87.2% | 89.2% | 91.2% |
| | | $\ell_2$ distance — 10 iter. (RQ) ↑ | 71.7% | 66.9% | 78.8% | 81.0% | 83.4% |
| | | $\ell_2$ distance — 30 iter. (RQ) ↑ | 44.7% | 33.7% | 51.5% | 47.9% | 46.1% |
| | | $\ell_2$ distance — 50 iter. (RQ) ↑ | 33.7% | 21.4% | 30.5% | 27.9% | 24.2% |
| | | $\ell_\infty$ distance — 5 iter. (RQ) ↑ | 76.4% | 76.1% | 88.7% | 90.2% | 91.8% |
| | | $\ell_\infty$ distance — 10 iter. (RQ) ↑ | 60.6% | 64.9% | 80.2% | 82.2% | 84.0% |
| | | $\ell_\infty$ distance — 30 iter. (RQ) ↑ | 29.5% | 33.0% | 53.1% | 48.0% | 47.0% |
| | | $\ell_\infty$ distance — 50 iter. (RQ) ↑ | 20.3% | 20.7% | 31.2% | 27.8% | 24.7% |
| | | Cosine similarity — 5 iter. ↓ | 0.99 | 0.26 | 0.06 | 0.06 | 0.07 |
| | | Cosine similarity — 10 iter. ↓ | 0.99 | 0.42 | 0.14 | 0.16 | 0.17 |
| | | Cosine similarity — 30 iter. ↓ | 1.00 | 0.90 | 0.63 | 0.77 | 0.80 |
| | | Cosine similarity — 50 iter. ↓ | 1.00 | 0.96 | 0.90 | 0.94 | 0.95 |
| Untargeted U-PGD attack | ε = 0.05 | $\ell_2$ distance — 5 iter. (UQ) ↓ | 3.2% | 91.1% | 1.9% | 0.4% | 0.1% |
| | | $\ell_2$ distance — 10 iter. (UQ) ↓ | 68.7% | 99.0% | 36.0% | 4.9% | 0.5% |
| | | $\ell_2$ distance — 30 iter. (UQ) ↓ | 99.9% | 99.8% | 99.3% | 98.7% | 96.5% |
| | | $\ell_2$ distance — 50 iter. (UQ) ↓ | 99.9% | 99.9% | 99.8% | 99.9% | 99.9% |
| | | $\ell_\infty$ distance — 5 iter. (UQ) ↓ | 8.7% | 88.2% | 1.7% | 0.4% | 0.1% |
| | | $\ell_\infty$ distance — 10 iter. (UQ) ↓ | 40.3% | 99.0% | 35.9% | 4.1% | 0.7% |
| | | $\ell_\infty$ distance — 30 iter. (UQ) ↓ | 99.9% | 99.9% | 99.1% | 98.3% | 95.5% |
| | | $\ell_\infty$ distance — 50 iter. (UQ) ↓ | 99.9% | 99.9% | 99.9% | 99.9% | 99.9% |
| | | Cosine similarity — 5 iter. ↑ | 0.99 | 0.22 | 0.72 | 0.74 | 0.80 |
| | | Cosine similarity — 10 iter. ↑ | 0.98 | 0.12 | 0.47 | 0.56 | 0.68 |
| | | Cosine similarity — 30 iter. ↑ | 0.26 | 0.07 | 0.16 | 0.10 | 0.16 |
| | | Cosine similarity — 50 iter. ↑ | -0.12 | 0.06 | 0.12 | 0.05 | 0.08 |
| | ε = 0.10 | $\ell_2$ distance — 5 iter. (UQ) ↓ | 58.9% | 91.6% | 3.4% | 0.8% | 0.1% |
| | | $\ell_2$ distance — 10 iter. (UQ) ↓ | 99.9% | 99.0% | 43.7% | 9.5% | 1.0% |
| | | $\ell_2$ distance — 30 iter. (UQ) ↓ | 99.9% | 99.8% | 99.5% | 99.3% | 98.9% |
| | | $\ell_2$ distance — 50 iter. (UQ) ↓ | 99.9% | 99.9% | 99.9% | 99.9% | 99.9% |
| | | $\ell_\infty$ distance — 5 iter. (UQ) ↓ | 50.5% | 92.3% | 3.4% | 0.8% | 0.2% |
| | | $\ell_\infty$ distance — 10 iter. (UQ) ↓ | 99.5% | 99.2% | 42.8% | 8.7% | 1.4% |
| | | $\ell_\infty$ distance — 30 iter. (UQ) ↓ | 99.9% | 99.9% | 99.4% | 99.2% | 98.4% |
| | | $\ell_\infty$ distance — 50 iter. (UQ) ↓ | 99.9% | 99.9% | 99.9% | 99.9% | 99.9% |
| | | Cosine similarity — 5 iter. ↑ | 0.99 | 0.16 | 0.66 | 0.71 | 0.78 |
| | | Cosine similarity — 10 iter. ↑ | 0.94 | 0.08 | 0.40 | 0.49 | 0.62 |
| | | Cosine similarity — 30 iter. ↑ | -0.06 | 0.03 | 0.13 | 0.08 | 0.12 |
| | | Cosine similarity — 50 iter. ↑ | -0.23 | 0.02 | 0.10 | 0.04 | 0.06 |
| | | Breakaway risk ↓ | 0.238% | 0.230% | 0.406% | 0.401% | 0.285% |
| | | Nearest neighbor accuracy ↑ | 0.0% | 0.0% | 0.0% | 0.0% | 0.0% |
| | | Overlap risk ↓ | 19.8% | 62.8% | 9.6% | 5.7% | 4.4% |
| | | Median adversarial margin ↑ | 11.1% | -3.4% | 18.0% | 19.6% | 26.6% |
| | | Average Certified Radius ↑ | 0.18 | 0.00 | 0.03 | 0.03 | 0.04 |
| Impersonation rate | | 3 iter. ↓ | 12.5% | 21.0% | 38.3% | 29.7% | 22.7% |
| | | 10 iter. ↓ | 43.2% | 52.4% | 77.5% | 75.1% | 76.0% |
| | | 30 iter. ↓ | 74.6% | 82.7% | 93.4% | 93.1% | 92.9% |
| | | 50 iter. ↓ | 83.9% | 90.2% | 95.6% | 95.4% | 95.4% |

ResNet50. MAE also has some of the best robustness against targeted U-PGD attacks and is competitive to PixPro for the untargeted case. It attains the lowest breakaway and overlap risks among all standard models as well as the largest median adversarial margin and is most robust to impersonation attacks.

In comparison, MOCOv3 scores rather poorly on most measures. Hence, the robustness of MAE cannot be solely attributed to the transformer backbone. This complements the observed variation in robustness performance among the ResNet50 models and provides evidence that it is likely the objective function, rather than the backbone architecture, that determines robustness.

The three adversarially trained MOCOv3 models witness a lower accuracy penalty than the corresponding MOCOv2 models: between 0.2% and 3.3% for the clean and between -0.9% and 0.8% for the lowpass accuracy for MOCOv3 compared with correspondingly 5.3%-10.2% and 2.8%-6.8% for MOCOv2. In fact, the adversarially trained MOCOv3 TAR model has a *higher* lowpass accuracy than the standard MOCOv3. Unsupervised adversarial training also leads to a uniformly better robustness against targeted and untargeted U-PGD attacks, albeit with a lower improvement compared to MO-COv2. We similarly witness large improvements in the overlap risk and median adversarial margin measures. The average certified radius and the certified accuracy (Fig. 10) are also significantly improved by adversarial training. Fig. 11 shows that the adversarially trained models are also more certifiably robust than the baseline MOCOv3.

However, the three adversarially trained models actually *fare worse* than the baseline MOCOv3 for breakaway risk and are *less robust* to impersonation attacks. The impersonation attacks are also less semantic in nature than the ones for the MOCOv2 adversarially trained models (Figs. 21 to 23 vs Figs. 26 to 28). This could also be due to the learning rate being too low, rather than due to transformer models being inherently more difficult to adversarially fine-tune in an unsupervised setting. The lower accuracy gap and the lower robustness further indicate that the adversarial training might not have been as "aggressive" for MOCOv3 as it was for MOCOv2. Still, while the improvements for MOCOv3 are not as drastic as for MOCOv2, unsupervised adversarial training does improve most robustness measures. The lower effectiveness for MOCOv3 of unsupervised adversarial training, especially in its role as a defence against impersonation attacks, is an avenue for future work that should examine whether there are fundamental differences between unsupervised adversarial training of CNN and Transformer models.

## C  TRAINING AND EVALUATION DETAILS

This section provides further details on the unsupervised adversarial training and the evaluation metrics implementations.

### C.1  UNSUPERVISED ADVERSARIAL TRAINING FOR MOCOV2

The three variants for the adversarially fine-tuned MOCOv2 are obtained by using a modification of the official MOCO source code. We perform fine-tuning by resuming the training procedure for additional 10 epochs but with the modified training loop. The unsupervised adversarial examples are concatenated to the model's $q$-inputs and the $k$-inputs are correspondingly duplicated as shown in List. 1. For $\bar{\mathcal{L}}$-PGD we use InfoNCE (Oord et al., 2018), the loss that MOCOv2 is trained with. All parameters, including the learning rate and its decay are as used for the original training and as reported by He et al. (2020). We only reduced the batch size from 256 to 192 in order to be able to train on four GeForce RTX 2080 Ti GPUs.

Listing 1: Pseudocode of adversarial fine-tuning for MoCo (modified from (He et al., 2020)).

```
# f_q, f_k: encoder networks for query and key
# queue: dictionary as a queue of K keys (CxK)
# m: momentum
# t: temperature

f_k.params = f_q.params # initialize

for x in loader: # load a minibatch x with N samples
    x_q = aug(x) # a randomly augmented version
```

```
    x_k = aug(x) # another randomly augmented version

    # perform the attack
    switch attack_type:
        case targeted:
            target_representation = roll(f_q(x_k), shifts=1)
            x_adv = targeted_upgd(x_q, target_representation)

        case untargeted:
            x_adv = untargeted_upgd(x_q)

        case loss:
            x_adv = batch_loss_upgd(f_q, x_q, f_k, x_k, queue, m, t)

    # get the representations
    q_clean = f_q.forward(x_q) # queries: NxC
    q_adv = f_q.forward(x_adv) # adversarial: NxC
    q = cat([q_clean, q_adv], dim=0)
    k = f_k.forward(x_k) # keys: NxC
    k = k.detach() # no gradient to keys

    # positive logits: 2Nx1
    l_pos = bmm(q.view(2*N,1,C), cat([k, k], dim=0).view(2*N,C,1))
    # negative logits: 2NxK
    l_neg = mm(q.view(2*N,C), queue.view(C,K))
    # logits: 2Nx(1+K)
    logits = cat([l_pos, l_neg], dim=1)
    # contrastive loss
    labels = zeros(2N) # positives are the 0-th
    loss = CrossEntropyLoss(logits/t, labels)
    # SGD update: query network
    loss.backward()
    update(f_q.params)
    # momentum update: key network
    f_k.params = m*f_k.params+(1-m)*f_q.params
    # update dictionary
    enqueue(queue, k) # enqueue the current minibatch
    dequeue(queue) # dequeue the earliest minibatch
```

## C.2 UNSUPERVISED ADVERSARIAL TRAINING FOR MOCOV3

Similarly to MOCOv2, the three variants for the adversarially fine-tuned MOCOv3 are obtained by using a modification of the official MOCOv3 source code. We perform fine-tuning by resuming the training procedure for additional 10 epochs but with the modified training loop. The unsupervised adversarial examples are added to the contrastive loss as shown in List. 2. All parameters, are as used for the original training and as reported by Chen et al. (2021). We only increased the learning rate from $1.5 \times 10^{-4}$ to $1.5 \times 10^{-3}$ and reduced the batch size from 256 to 192 in order to be able to train on four GeForce RTX 2080 Ti GPUs.

Listing 2: Pseudocode of adversarial fine-tuning for MOCOv3.

```
# f_base: base encoder network
# f_predictor: predictor network
# m: momentum

f_momentum.params = f_base.params # initialize momentum encoder

for x in loader: # load a minibatch x with N samples
    x_0 = aug(x) # a randomly augmented version
    x_1 = aug(x) # another randomly augmented version

    # perform the attack
    switch attack_type:
```

```
        case targeted:
            target_representation = roll(f_predictor(f_base(x_1)), shifts=1)
            x_adv = targeted_upgd(x_0, target_representation)

        case untargeted:
            x_adv = untargeted_upgd(x_0)

        case loss:
            x_adv = batch_loss_upgd(f_base, f_predictor, x_0, x_1)

    # update the momentum encoder
    f_momentum.params = f_momentum.params * m + f_base.params * (1-m)

    # get the base representations
    q_0 = f_predictor.forward(f_base.forward(x_0)) # x_0 reps: NxC
    q_1 = f_predictor.forward(f_base.forward(x_1)) # x_1 reps: NxC
    q_adv = f_predictor.forward(f_base.forward(x_adv)) # attacked reps: NxC

    # get the momentum representations
    k_0 = f_momentum.forward(x_0) # x_0 reps: NxC
    k_1 = f_momentum.forward(x_1) # x_1 reps: NxC
    k_adv = f_momentum.forward(x_adv) # attacked reps: NxC

    # compute the loss
    loss = contrastive_loss(q_0, k_1) + contrastive_loss(q_1, k_0) + \
           contrastive_loss(q_adv, k_1) + contrastive_loss(q_1, k_adv)

    # parameter update: query network
    loss.backward()
    update(f_q.params)
```

### C.3 Unsupervised adversarial training for ResNet

The two unsupervised adversarially fine-tuned ResNet50 models are obtained by using an implementation similar to the one for MOCOv2 and MOCOv3. However, as the ResNet model is trained in a supervised setting there is no natural contrastive loss to use. That is why we use the MSE loss on the representations. For the targeted case, we also ensure that the clean samples are still mapped to the original representations. The full procedure is outlined in List. 3.

Listing 3: Pseudocode of adversarial fine-tuning for ResNet.

```
# f_base: base encoder network
# f_finetuned: the finetuned network

f_finetuned.params = f_base.params # initialize momentum encoder

for x in loader: # load a minibatch x with N samples
    x = aug(x) # a randomly augmented batch

    clean_reps = f_base(x)

    # perform the attack
    switch attack_type:
        case targeted:
            target_representation = roll(clean_reps, shifts=1)
            x_adv = targeted_upgd(x, target_representation)
            loss = MSELoss(cat([clean_reps, clean_reps], axis=0),
                        f_funetuned(cat([x, x_adv], axis=0)))

        case untargeted:
            x_adv = untargeted_upgd(x)
            loss = MSELoss(clean_reps, f_funetuned(x_adv))
```

```
# parameter update
loss.backward()
update(f_finetuned.params)
```

### C.4 LINEAR PROBES

As part of the evaluation we train three linear probes for each model.

- **Standard linear probe:** for computing the top-1 and top-5 accuracy on clean samples, as well as for the impersonation attack evaluation.
- **Lowpass linear probe:** for computing the top-1 and top-5 accuracy on samples with removed high-frequency components. We use the implementation of Wang et al. (2020) and keep only the Fourier components that are within a radius of 50 from the center of the Fourier-transformed image.
- **Gaussian noise linear probe:** trained on samples with added Gaussian noise for computing the certified accuracy as randomized smoothing results in a more robust model when the base model is trained with aggressive Gaussian noise (Lecuyer et al., 2019). Therefore, we add Gaussian noise with $\sigma = 0.25$ to all inputs.

All linear probes are trained with the train set of ImageNet Large Scale Visual Recognition Challenge 2012 (Russakovsky et al., 2015) and are evaluated on its test set. For training we use modification of the MOCO linear probe evaluation code for 25 epochs. The starting learning rate is 30.0 with 10-fold reductions applied at epochs 15 and 20.

For fairness of the comparison, we use the same implementation to evaluate all models. Therefore, there might be differences between the accuracy values reported by us and the ones reported in the original publications of the respective models.

### C.5 COMPUTING THE DISTRIBUTION OF INTER-REPRESENTATIONAL DIVERGENCES

The distribution of $\ell_2$ and $\ell_\infty$-induced divergences between the representations of clean samples of PASS (Asano et al., 2021) is needed for computing the universal and relative quantiles. Due to computational restrictions, we compute the representations of 10,000 samples and the divergences between all pairs of them in order to construct the empirical estimate of the distribution of inter-representational divergences. We observe that 10,000 samples are more than sufficient for the empirical estimate of the distribution to converge.

### C.6 ADVERSARIAL ATTACKS

In Tabs. 2, 3 and 5 to 7 we report U-PGD attacks performed with $d(x, x') = \|x - x'\|_2$ and $\alpha = 0.001$. We report median values over the same 1000 samples from PASS (Asano et al., 2021). The median universal quantile is reported for targeted attacks and the median relative quantile is reported for targeted attacks as explained in Sec. 5. In Tabs. 2 and 3 we report only the resulting $\ell_2$ quantiles, while in the extended results (Tabs. 5 to 7) we also show the $\ell_\infty$ quantiles and cosine similarities.

### C.7 BREAKAWAY RISK AND NEAREST NEIGHBOR ACCURACY

The breakaway risk and nearest neighbor accuracy are also computed by attacking the same 1000 samples from PASS ($D'$) and computing their divergences with all other samples from PASS ($D$). Our empirical estimate is then :

$$\hat{p}_{\text{breakaway}} = \frac{1}{|D'|(|D| - 1)} \sum_{x_i \in D'} \sum_{x_j \in D/\{x_i\}} \mathbb{1}\left[d(f(\hat{x}_i), f(x_j)) < d(f(\hat{x}_i), f(x_i))\right],$$

where $\hat{x}_i$ is the untargeted U-PGD attack with $d(x, x') = \|x - x'\|_2$, $\epsilon = 0.05$ and $\alpha = 0.001$ for 25 iterations.

## C.8 OVERLAP RISK AND MEDIAN ADVERSARIAL MARGIN

The overlap risk and median adversarial margin are computed over 1000 pairs of samples from PASS ($D'$). Each element of the pair is attacked to have a representation similar to the other element.

$$\hat{p}_{\text{overlap}} = \frac{1}{|D'|} \sum_{(x,x') \in D'} \mathbb{1}\left[ d(f(x_i), f(\hat{x}_j^{\rightarrow i})) < d(f(x_i), f(\hat{x}_i^{\rightarrow j})) \right],$$

where $\hat{x}_i^{\rightarrow j}$ is the targeted U-PGD attack on $x_i$ towards $x_j$ with $d(x, x') = \|x - x'\|_2$, $\epsilon = 0.05$ and $\alpha = 0.001$ for 10 iterations.

## C.9 CERTIFIED ACCURACY

We use the randomized smoothing implementation by Cohen et al. (2019). We evaluate the Gaussian noise linear probe (see App. C.4) over 200 samples from the ImageNet test set (Russakovsky et al., 2015). We use $\sigma = 0.25$, $N_0 = 100$, $N = 100,000$ and an error probability $\alpha = 0.001$, as originally used by Cohen et al. (2019). Figs. 3 and 10 show the resulting certified accuracy for MOCOv2, MOCOv3, and their unsupervised adversarially trained versions. Fig. 6 shows the certified accuracy for the other models. These plots show the fraction of samples which are correctly classified and which certifiably have the same classification within a given $\ell_2$ radius of the input space.

Tabs. 5 to 7 also show the Average Certified Radius for all models. The Average Certified Radius was proposed by Zhai et al. (2020) as a way to summarize the certified accuracy vs radius plots with a single number. The average certified radius can be computed as:

$$\text{ACR} = \sum_{x \in D} \mathbb{1}[\text{correct}(x)] \cdot \text{certified\_radius}(x).$$

## C.10 CERTIFIED ROBUSTNESS

The certified robustness evaluation in Figs. 4 and 10 was done with the center smoothing implementation by Kumar & Goldstein (2021). We evaluate the models over the same 200 samples from the ImageNet test set (Russakovsky et al., 2015). We use $\sigma = 0.25$, $N_0 = 10,000$, $N = 100,000$ and error probabilities $\alpha_1 = 0.005$, $\alpha_2 = 0.005$, as originally proposed by Kumar & Goldstein (2021).

## C.11 IMPERSONATION ATTACKS

The impersonation attack evaluation is performed using targeted U-PGD attacks. We use the Assira dataset that contains 25,000 images, equally split between cats and dogs (Elson et al., 2007). When evaluating a model, we consider only the subset of images that the standard linear probe (see App. C.4) for the given model classifies correctly. Then, we construct pairs of an image of a cat and an image of a dog. We perform two attacks: attacking the cat to have a representation as close as possible to that of the dog and vice-versa. The attacked images are then classified with the clean linear probe. The success rate of cats impersonating dogs and dogs impersonating cats are computed separately and then averaged to account for possible class-based differences. Note that the linear probe is *not* used for constructing the attack, i.e. we indeed fool it without accessing it. App. D shows examples of the impersonation attacks for all models.

## D IMPERSONATION ATTACKS

This appendix showcases samples of the impersonation attacks on the models discussed in the paper. The first and third row in each sample are the original images of cats and dogs respectively. The second row is the result when each cat image is attacked to have a representation close to the representation of the corresponding dog image. The fourth row is the opposite: the dog image attacked to have a representation close to the representation of the cat image. The attack used was targeted U-PGD with $d(r, r') = \|r - r'\|_2$ for 50 iterations with $\epsilon = 0.10$ and $\alpha = 0.01$. The samples shown differ from model to model as we restrict the evaluation to the samples that are correctly predicted by the given model, see App. C.11 for details.

A key observation is that the perturbations necessary to fool the standard models visually appear as noise (Figs. 12 and 16 to 20). However, the perturbations applied to the three adversarially trained MOCOv2 models (Figs. 21 to 23), as well as the supervised adversarially trained ResNet50 (Fig. 13), are more "semantic" in nature, and in some cases even resemble features of the target class. Still, this is not the case when comparing the impersonation attacks on MOCOv3 (Fig. 25) with the attacks on the adversarially trained versions (Figs. 26 to 28).

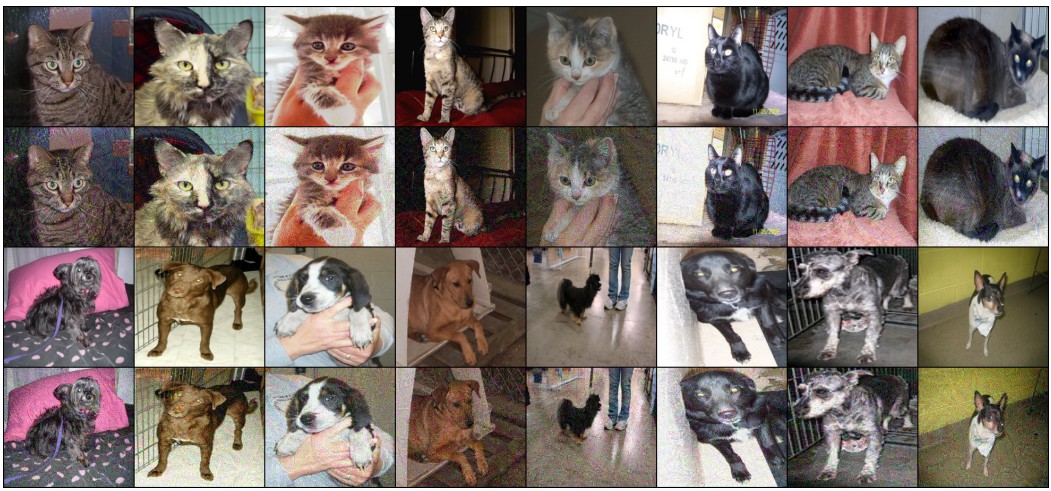

Figure 12: Impersonation attack samples for ResNet50 (He et al., 2016).

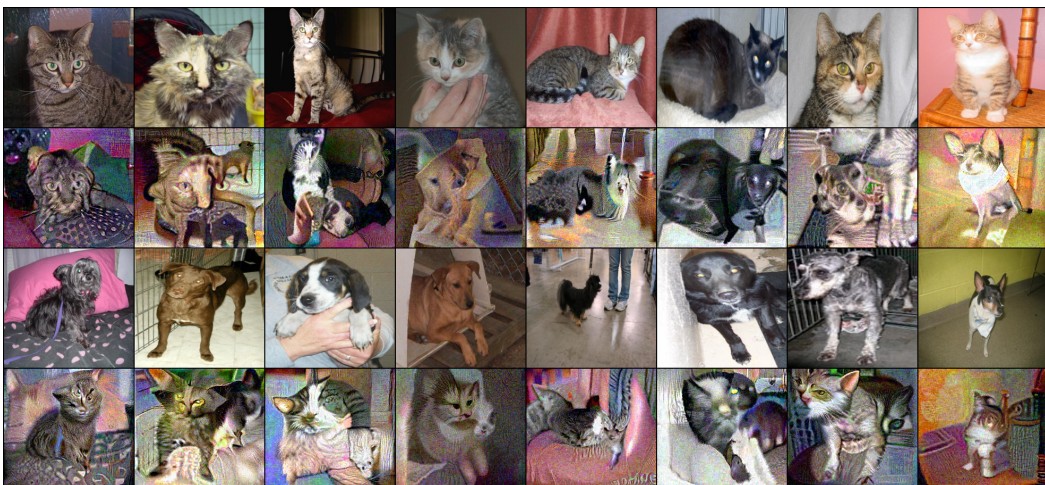

Figure 13: Impersonation attack samples for supervised adversarially trained ResNet50 (Salman et al., 2020).

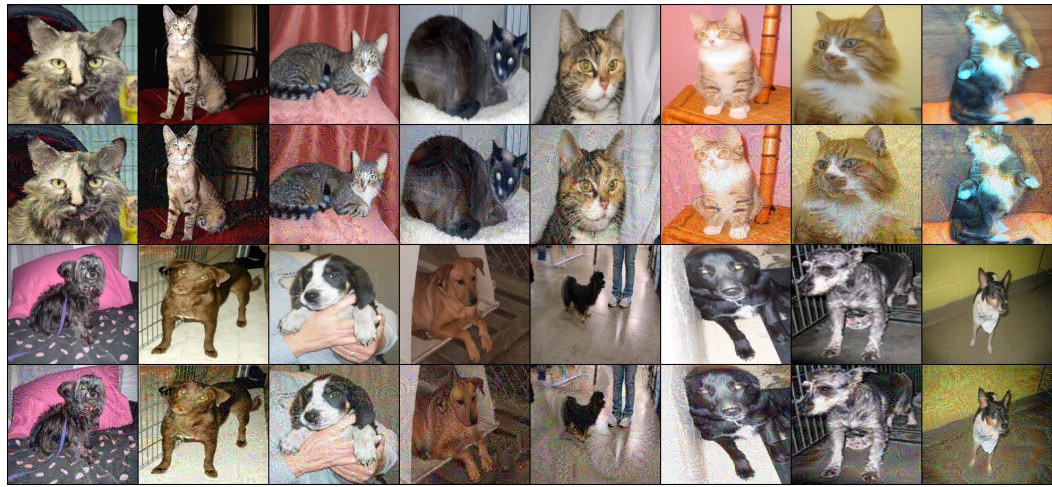

Figure 14: Impersonation attack samples for ResNet50 TAR.

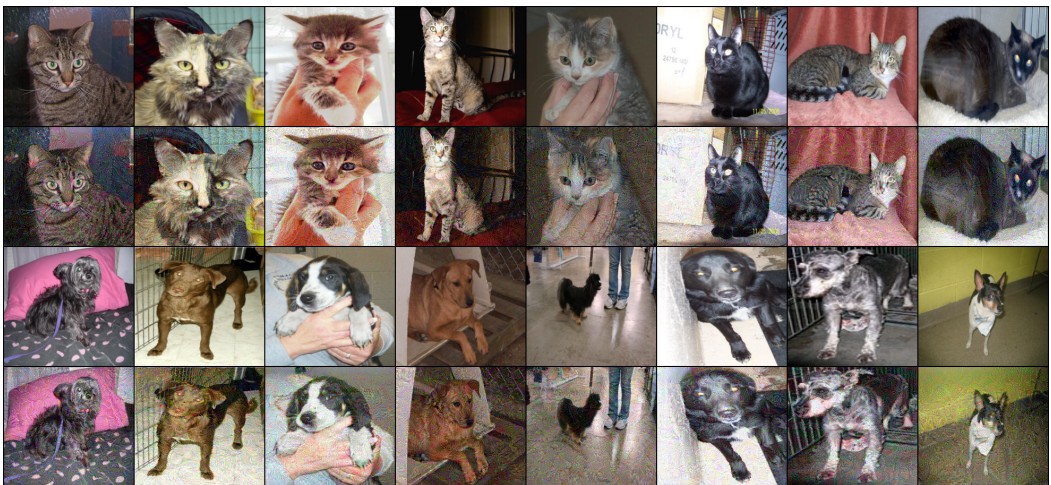

Figure 15: Impersonation attack samples for ResNet50 UNTAR.

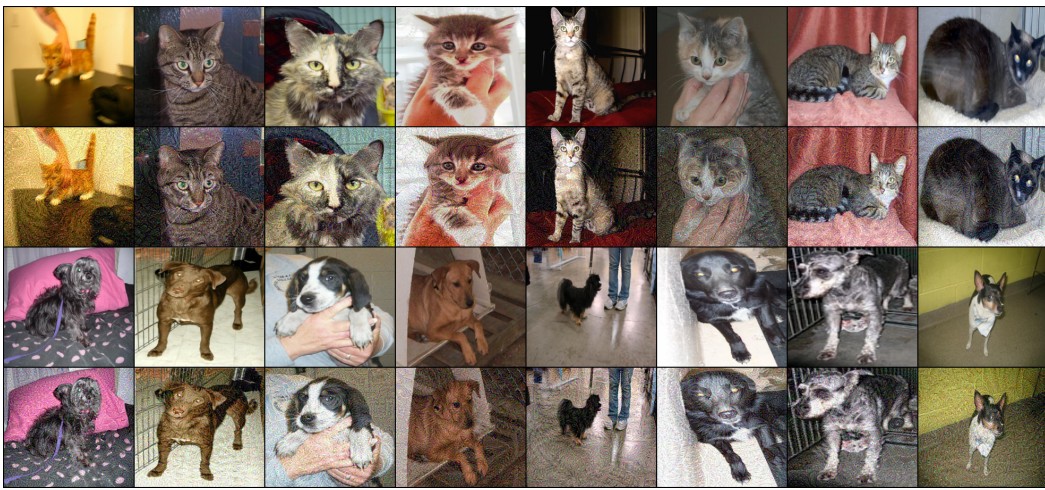

Figure 16: Impersonation attack samples for MOCO with non-semantic negatives (Ge et al., 2021).

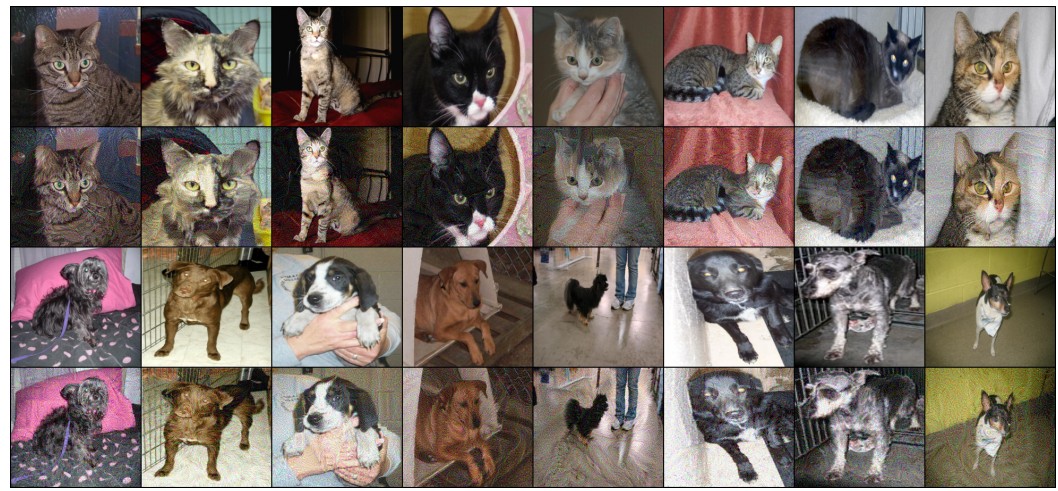

Figure 17: Impersonation attack samples for PixPro (Xie et al., 2021).

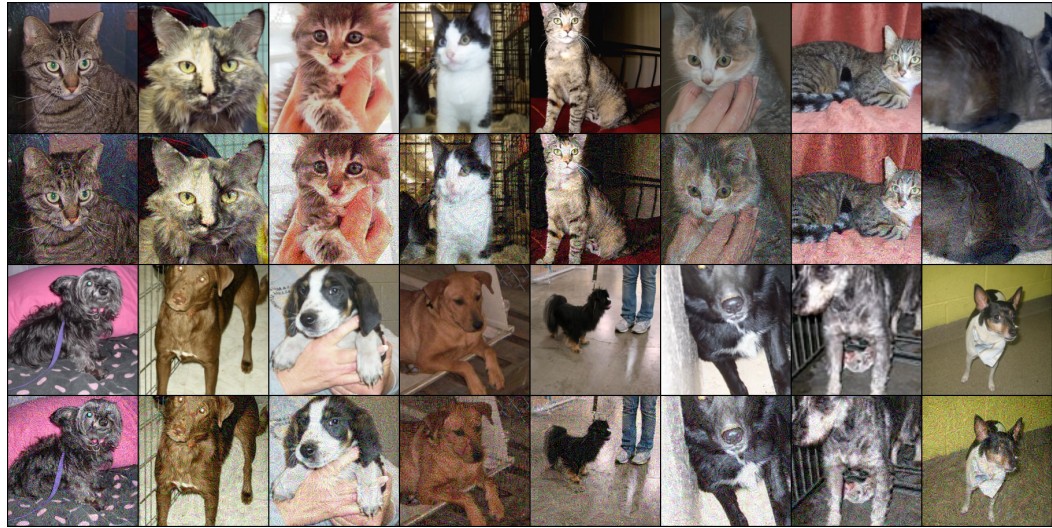

Figure 18: Impersonation attack samples for SimCLRv2 (Chen et al., 2020c).

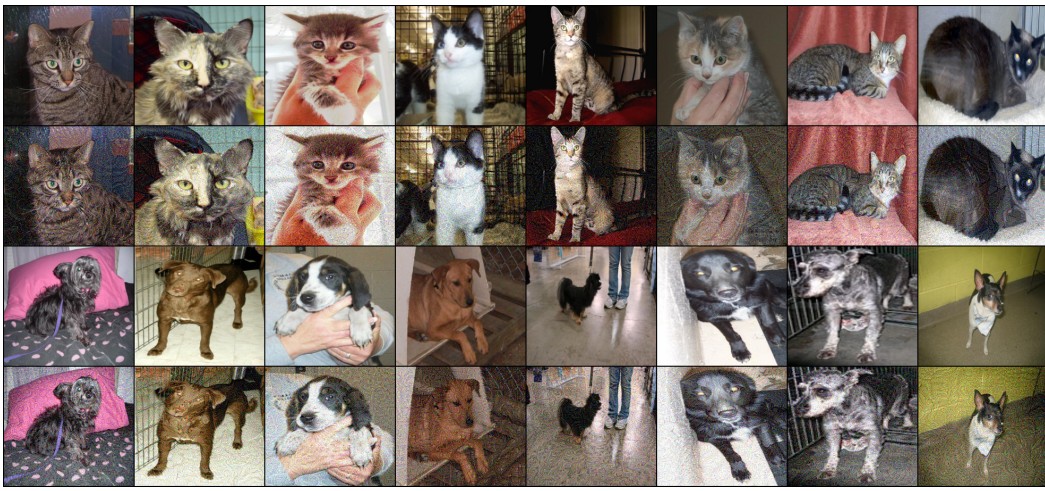

Figure 19: Impersonation attack samples for SimSiam (Chen & He, 2021).

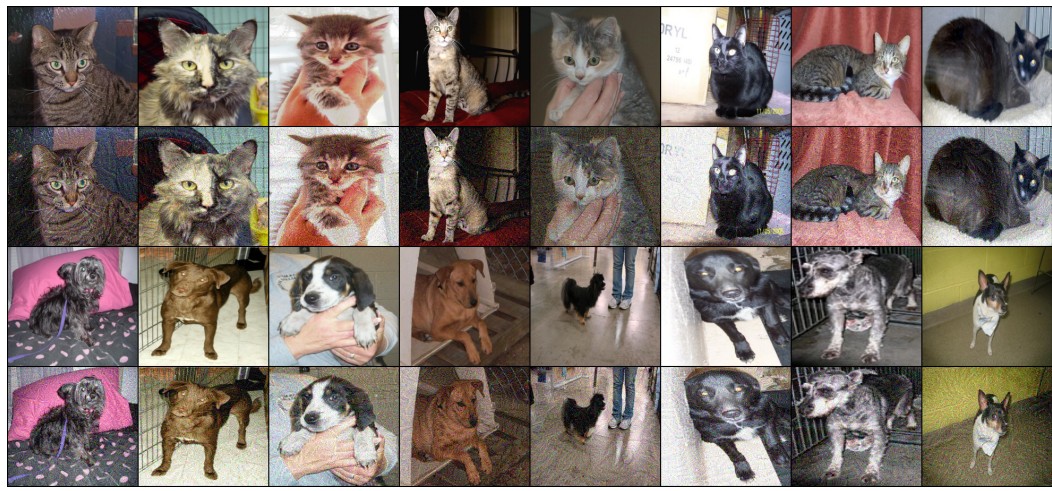

Figure 20: Impersonation attack samples for MOCOv2 (He et al., 2020; Chen et al., 2020d).

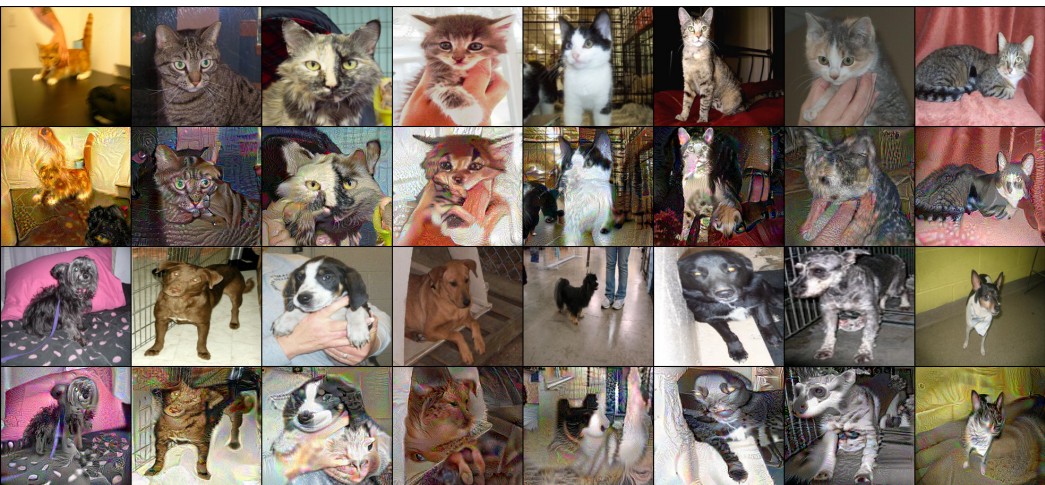

Figure 21: Impersonation attack samples for MOCOv2 TAR.

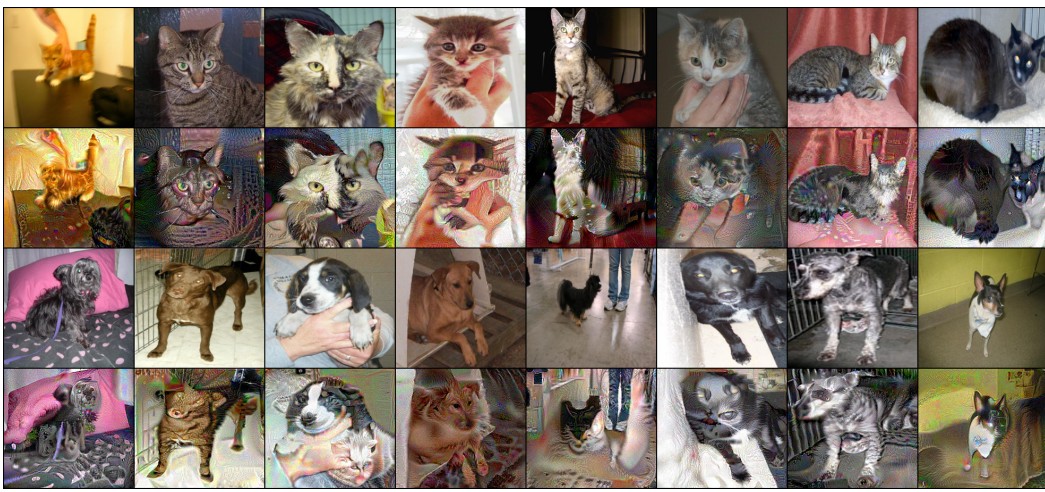

Figure 22: Impersonation attack samples for MOCOv2 UNTAR.

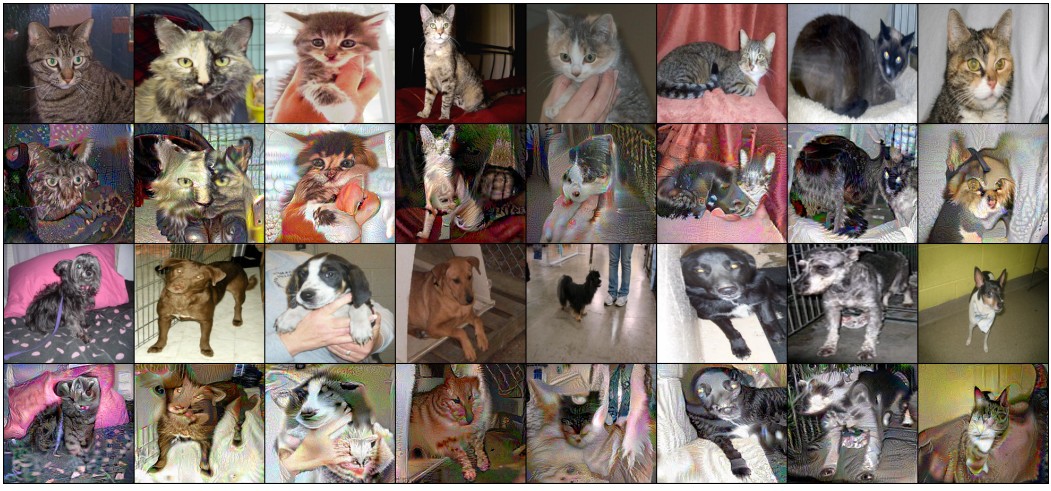

Figure 23: Impersonation attack samples for MOCOv2 LOSS.

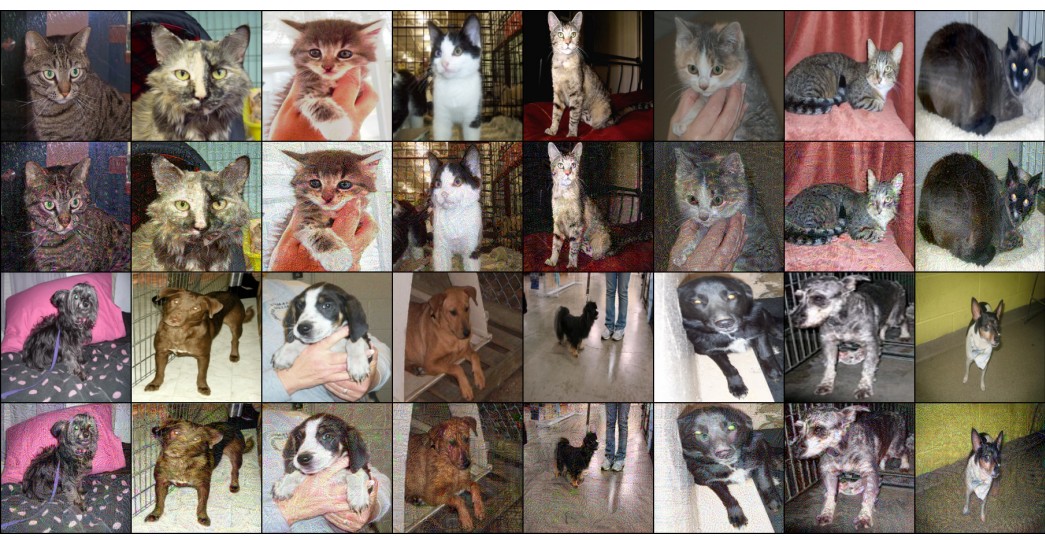

Figure 24: Impersonation attack samples for MAE (He et al., 2022).

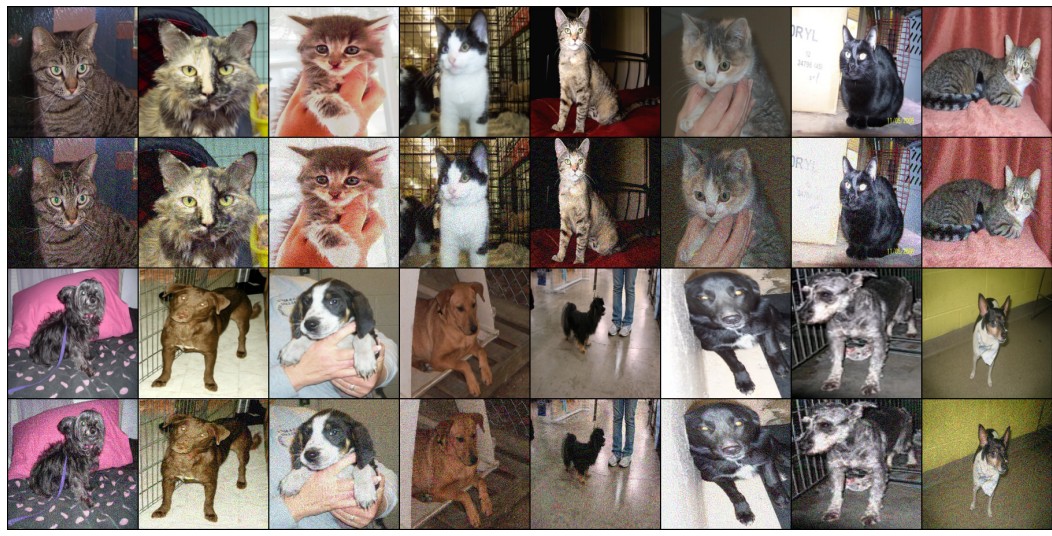

Figure 25: Impersonation attack samples for MOCOv3 (Chen et al., 2021).

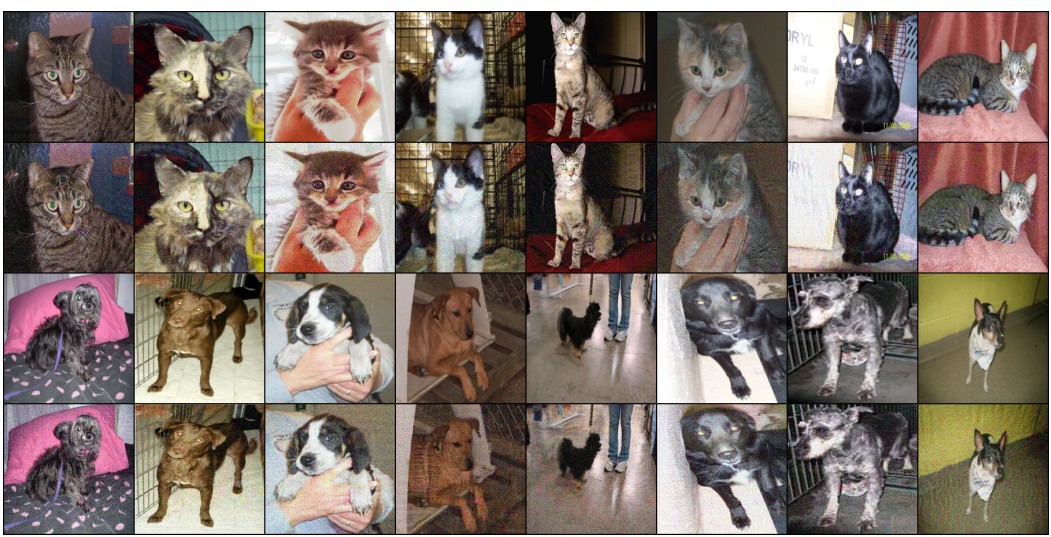

Figure 26: Impersonation attack samples for MOCOv3 TAR.

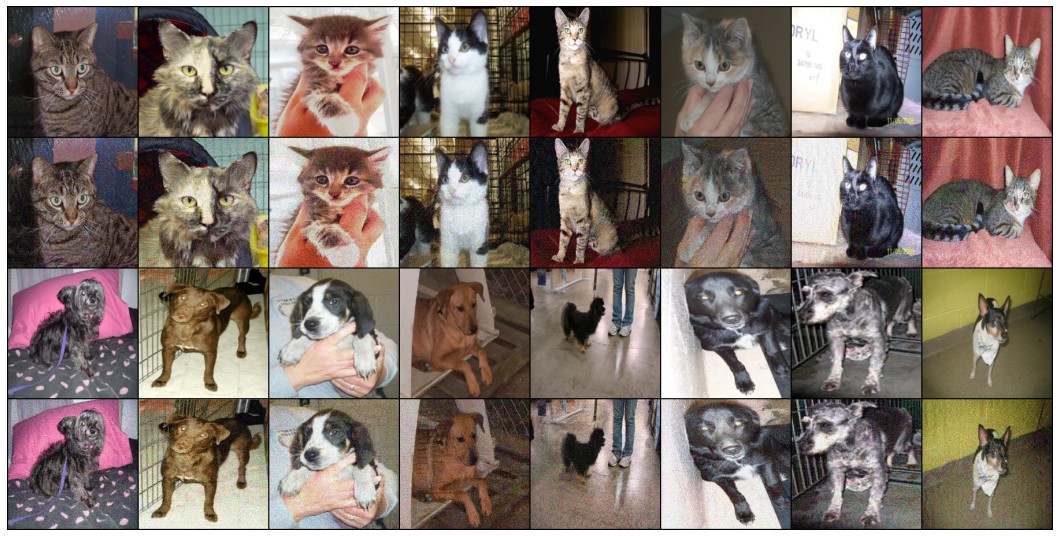

Figure 27: Impersonation attack samples for MOCOv3 UNTAR.

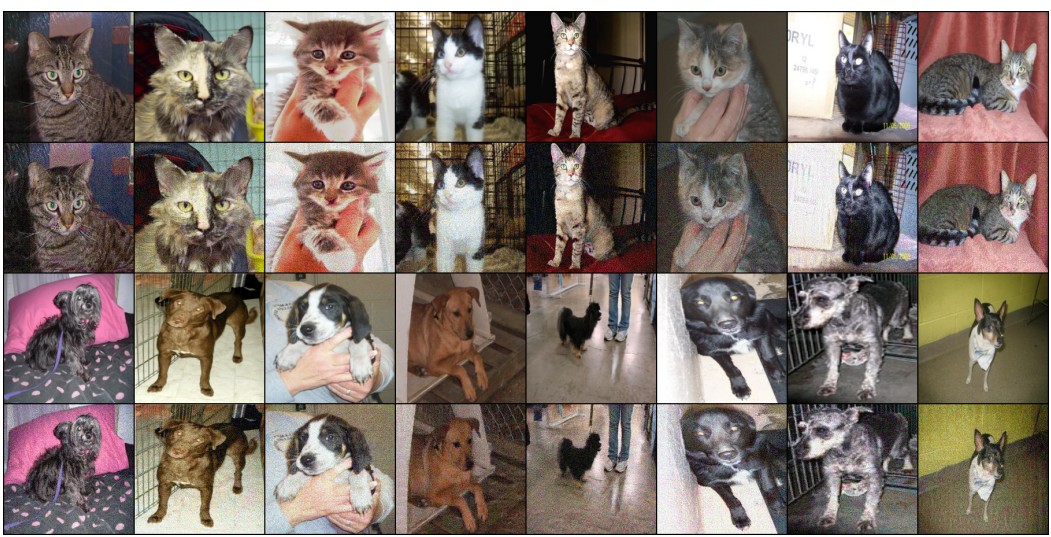

Figure 28: Impersonation attack samples for MOCOv3 LOSS.

