# OpenReview forum: "Robustness of Unsupervised Representation Learning without Labels"
_ICLR.cc/2023/Conference — Submitted to ICLR 2023_

### Official Review · Reviewer_HYfP · 2022-10-20

**Confidence:** 4
**Correctness:** 4
**Technical Novelty And Significance:** 4
**Empirical Novelty And Significance:** 4
**Recommendation:** 8

**Clarity, Quality, Novelty And Reproducibility:**

Overall, I think the writing and presentation is very clear.  I also think that the work is novel as it is the first work to rigorously define what properties we would expect a robust encoder to follow and propose metrics for measuring the robustness of an encoder.  Overall, I think that the metrics are well-motivated and can be very useful for assessing the robustness of current training techniques for adversarial ML.

**Strength And Weaknesses:**

Strengths:
- Definitions for breakaway risk and overlap risk are well-motivated and easy to understand.
- Interesting problem setting (measuring robustness without labels)
- Well-motivated metrics
- Show that fine tuning based on some of the proposed metrics can greatly improve robustness of encoder
- Writing is clear

Weaknesses:
- The proposed metrics could be used to analyze encoders trained using any technique: supervised learning, self-supervised learning, and unsupervised learning, but the experiments are restricted to unsupervised encoders.  I think it would be interesting to perform evaluations on encoders of adversarially trained models (supervised learning but drop the final classifier layer) and see if these learned representations actually satisfy the definitions proposed.  Additionally, it would be interesting to see whether the pretraining approaches proposed by works for adversarially robust self-supervised learning (ie. Kim et al 2020) lead to more robust encoders.

**Summary Of The Paper:**

In this work, the authors define what it means for an encoder to be robust and describe metrics for measuring robustness.  Using these metrics, they then evaluate existing SOTA representation encoders with and without adversarial finetuning (based on the proposed metrics).

**Summary Of The Review:**

I find this paper very interesting as it rigorously defines what it means for an encoder to be robust, and I think that the proposed metrics will be very useful to the adversarial ML research community especially since recently there have been works proposing methods for robust pretraining.  I am a little disappointed by the experiments though, not that they are bad or uninteresting, I just feel like the authors introduced a very powerful toolkit but did not use this toolkit to its full potential.

---

> ### Author Response · Authors · 2022-11-11
> **Initial response to comments**
>
> > The proposed metrics could be used to analyze encoders trained using any technique: supervised learning, self-supervised learning, and unsupervised learning, but the experiments are restricted to unsupervised encoders. [...] Additionally, it would be interesting to see whether the pretraining approaches proposed by works for adversarially robust self-supervised learning (ie. Kim et al 2020) lead to more robust encoders.
>
> We agree with these suggestions. Indeed further experiments with more models could only further strengthen our claims. However, these experiments are quite costly to run. While the adversarial fine-tuning and evaluation themselves are rather computationally cheap, training the linear probes is quite expensive and there are 3 per model. We, however, appreciate your suggestion and will try to add experiments with a supervised encoder.
>
> > I think it would be interesting to perform evaluations on encoders of adversarially trained models (supervised learning but drop the final classifier layer) and see if these learned representations actually satisfy the definitions proposed.
>
> This is a great suggestion! We evaluated the penultimate layer of a (supervised) adversarially trained ResNet50 model in the exact same way as we did for the standard ResNet50. The adversarially trained ResNet50 was significantly more robust than the standard ResNet50 on every single measure, almost maxing out many of the measures. This demonstrates that our measures successfully detect models which we know to be robust in a supervised setting and further strengthens our methods and results. We will add this to the revised manuscript. Thank you for the suggestion!
>
> |                                        | ResNet50 | Robust ResNet50 |
> |---------------------------------------:|---------:|----------------:|
> |   Targeted U-PGD, ɛ = 0.05,  5 iter. ↑ | 66.38%   | 98.14%          |
> |   Targeted U-PGD, ɛ = 0.05, 10 iter. ↑ | 52.68%   | 96.45%          |
> |   Targeted U-PGD, ɛ = 0.05, 50 iter. ↑ | 27.42%   | 86.89%          |
> |   Targeted U-PGD, ɛ = 0.10,  5 iter. ↑ | 69.27%   | 97.92%          |
> |   Targeted U-PGD, ɛ = 0.10, 10 iter. ↑ | 55.63%   | 96.03%          |
> |   Targeted U-PGD, ɛ = 0.10, 50 iter. ↑ | 27.92%   | 83.90%          |
> | Untargeted U-PGD, ɛ = 0.05,  5 iter. ↓ | 98.70%   | 0.00%           |
> | Untargeted U-PGD, ɛ = 0.05, 10 iter. ↓ | 99.90%   | 0.00%           |
> | Untargeted U-PGD, ɛ = 0.05, 50 iter. ↓ | 99.90%   | 0.01%           |
> | Untargeted U-PGD, ɛ = 0.10,  5 iter. ↓ | 99.40%   | 0.00%           |
> | Untargeted U-PGD, ɛ = 0.10, 10 iter. ↓ | 99.90%   | 0.00%           |
> | Untargeted U-PGD, ɛ = 0.10, 50 iter. ↓ | 99.90%   | 0.90%           |
> |                       Breakaway risk ↓ | 0.120%   | 0.000%          |
> |                Nearest neighbor acc. ↑ | 0.00%    | 95.00%          |
> |                         Overlap risk ↓ | 92.48%   | 0.00%           |
> |              Med. adversarial margin ↑ | -18.57%  | 84.86%          |
> |                Avg. Certified Radius ↑ | 0.39     | 0.77            |
> |           Impersonation rate 3 iter. ↓ | 41.49%   | 6.38%           |
> |          Impersonation rate 10 iter. ↓ | 76.50%   | 49.92%          |
> |          Impersonation rate 50 iter. ↓ | 89.94%   | 86.90%          |

---

> > ### Comment · Reviewer_HYfP · 2022-12-09
> > **Thank you for the response**
> >
> > Thank you for the additional experiments with standard trained vs adversarially trained ResNet50.  I think this strongly demonstrates that the proposed metrics for unsupervised robustness are also to some degree aligned with supervised notions of robustness.  I also read through other reviews and think that the authors have addressed most concerns.  Because of this, I will increase my score to 8.

---

### Official Review · Reviewer_TA8W · 2022-10-24

**Confidence:** 4
**Correctness:** 2
**Technical Novelty And Significance:** 2
**Empirical Novelty And Significance:** 2
**Recommendation:** 3

**Clarity, Quality, Novelty And Reproducibility:**

The organization of this paper is not very good. It is somewhat weird to suddenly illustrate adversarial training which is a defensive strategy in Section 4 of unsupervised attacks. This paper proposes two unsupervised adversarial risks and realizes them based on PGD and conventional distance metrics. But it seems that the metric is not consistent. Therefore, the technical quality seems to be not very good and the novelty is somewhat fair. The author provides algorithm and experimental details. Thus, this paper has good reproducibility.

**Strength And Weaknesses:**

Strength
+ This paper proposes two interesting unsupervised adversarial risks based on the distance between benign and adversarial data. The risks indeed solve the problem that what value of the distance being “small”.
+ This paper gives several robustness metrics, including quantiles for (un)targeted attacks, estimation of breakaway risk and overlap risk, nearest neighbour accuracy, adversarial margin, and certified robustness.

Weaknesses:
- This paper seems to have minor technical novelty. The main techniques of adversarial attacks and adversarial training are based on PGD or FGSM and even the loss function is simply the conventional distance.
- The relationship (such as differences and similarities) between breakaway risk and overlap risk is not very clear. I am confused about which risk really evaluates unsupervised adversarial robustness.
- It is somewhat difficult to understand how the metric “universal quantiles for untargeted attacks” can evaluate robustness. This metric closely depends on the sampling procedure. It is hard to say a model is non-robust if the fraction of distance between $x’$ and $x’’$ is smaller than the distance between $\hat{x}$ and $x$ is smaller since it could be incurred by the sampling procedure.
- The claim that “the breakaway risk can be very small” is somewhat confusing. Due to this claim, the author proposes nearest neighbour accuracy. However, Table 2 seems to show that breakaway risk can better differentiate the robustness of each model than nearest neighbour accuracy since nearest neighbour accuracy is almost very low and the same among different models.
- It seems that different unsupervised adversarial metrics do not provide a consistent evaluation. For example, in Table 2, ResNet50 has a lower breakaway risk while a higher overlap risk and PixPro has a higher breakaway risk while a lower overlap risk. Therefore, it is hard to compare the unsupervised robustness between these models using these two proposed metrics.
- It is not very clear the relationship between unsupervised robustness evaluation and supervised robustness evaluation. Will the unsupervised robust accuracy and supervised robust accuracy be positively correlated between them?


**Summary Of The Paper:**

This paper proposes two unsupervised adversarial risks, i.e., breakaway risk and overlap risk, that evaluate the adversarial robustness without requiring labels. Further, this paper generates unsupervised adversarial attacks via FGSM and PGD maximizes or minimizes the distance between benign and adversarial data. The authors also propose to use adversarial training based on the unsupervised FGSM or PGD to improve the robustness. Then, this paper provides a series of robustness measurements and empirically shows that adversarial-trained models are more robust than standard-trained models.

**Summary Of The Review:**

This paper proposes two unsupervised adversarial risks and realizes them based on PGD and conventional distance metrics. However, I have several concerns that have been illustrated in Weaknesses section to be solved.

---

> ### Author Response · Authors · 2022-11-11
> **Initial response to comments (Part I)**
>
> > This paper seems to have minor technical novelty. The main techniques of adversarial attacks and adversarial training are based on PGD or FGSM and even the loss function is simply the conventional distance.
>
> We agree that the main techniques of adversarial attacks and adversarial training are based on the well-known PGD and FGSM attacks. However, in our view, the simplicity of how a small modification of two popular supervised attacks can unlock methods for unsupervised robustness is one of the strong contributions of our work. This means that our methods are simple to understand, easy to implement, and can benefit from a large body of prior work studying the properties of the PGD and FGSM attacks.
>
> The loss function is not simply the conventional distance. For example, our L-PGD models are trained with a complex contrastive objective. Furthermore, in Appendix A we illustrate how our method can be also used with the KL divergence, Wasserstein distance and various contrastive losses.
>
> While we believe that extending existing attacks to the unsupervised setting is in itself a technical novelty, the novelty of our work is not limited to this contribution. In particular, to the best of our knowledge, we are the first to propose looking at unsupervised robustness as a desirable property in itself without the context of specific tasks and to define measures to assess it, such as our breakaway and overlap risks. We are also the first to propose a method for certified assessment of unsupervised robustness.
>
>
> > The relationship (such as differences and similarities) between breakaway risk and overlap risk is not very clear. I am confused about which risk really evaluates unsupervised adversarial robustness.
>
> In contrast to the supervised setting, there is no clear single way to evaluate unsupervised adversarial robustness. This is one of the main points we are trying to communicate in this work, as we discuss in Sections 7 and 8. Therefore, both the breakaway risk and the overlap risk are candidate measures for unsupervised adversarial robustness. Furthermore, in our conclusion, we invite the community to propose and study other measures as well.
>
> Both risk measures we propose aim to assess how successful the encoder is at concentrating perturbed samples closely together. However, they look at this objective from two different perspectives: the breakaway risk essentially looks at what is the likelihood that a downstream instance classifier that works as a nearest neighbour classifier misclassifies a perturbed sample as being generated from the wrong clean sample. Therefore, it measures whether parts of the perturbed space of a sample are mapped to regions of the representation space which are very far (broken away) from the representation of the clean sample. The overlap risk, on the other hand, assesses the probability that parts of the representation space are shared by perturbations of two different samples. If that was the case, then no downstream task would be able to distinguish between the two samples as they would be mapped to the same part of the representation space.
>
> Therefore, both the breakaway risk and the overlap risk measure unsupervised adversarial robustness, albeit two separate takes on it.
>
> > It is somewhat difficult to understand how the metric “universal quantiles for untargeted attacks” can evaluate robustness. This metric closely depends on the sampling procedure. It is hard to say a model is non-robust if the fraction of distance between $x'$ and $x''$ is smaller than the distance between $\hat x$ and $x$ is smaller since it could be incurred by the sampling procedure.
>
> By “sampling procedure” in your comment we understand the method to pick the pairs $(x', x'')$ used to obtain the empirical estimates for the two risks, as well as the pairs used to compute the empirical distribution of divergences between representations. And you are right that the sampling procedure can result in different distributions. In our work, we take the most agnostic approach to the sampling procedure and we pick samples at random from the dataset which is the empirical equivalent to sampling from the underlying data distribution. Our measures are always defined relative to this distribution as shown in Equations 1 and 2. However, this is not a limitation of our work as robustness evaluation is almost always restricted to a specific dataset. Still, we will try to stress the dependence on the underlying distribution and dataset more explicitly in the revised manuscript.

---

> ### Author Response · Authors · 2022-11-11
> **Initial response to comments (Part II)**
>
> > The claim that “the breakaway risk can be very small” is somewhat confusing. Due to this claim, the author proposes nearest neighbour accuracy. However, Table 2 seems to show that breakaway risk can better differentiate the robustness of each model than nearest neighbour accuracy since nearest neighbour accuracy is almost very low and the same among different models.
>
> Our claim is actually that “the breakaway risk can be very small for __robust__ encoders”. You are absolutely right that in Table 2, the breakaway risk differentiates better between the standard encoders. However, when comparing the three adversarially trained encoders in Table 3, they all have very low breakaway risk (0.049%, 0.001%, and 0.000%) but a much larger range of nearest neighbour accuracies (17%, 35.6%, and 64.4%). This difference indeed strengthens our claims that a suite of measures is necessary to evaluate unsupervised robustness as it is a multifaceted problem.
>
> > It seems that different unsupervised adversarial metrics do not provide a consistent evaluation. For example, in Table 2, ResNet50 has a lower breakaway risk while a higher overlap risk and PixPro has a higher breakaway risk while a lower overlap risk. Therefore, it is hard to compare the unsupervised robustness between these models using these two proposed metrics.
>
> As in our above comment, the fact that different measures do not provide a consistent evaluation is one of our key findings. The difference between the models in Table 2 is the focus of the “There is no most robust standard model” paragraph in the Results section. We also comment on this in the Conclusion: _“there is no single unsupervised robustness measure: models can have drastically different performance across the different metrics”_.
>
> It is indeed hard to compare the models when their robustness is not consistent across the different measures but this isn’t an issue with our work but rather highlights that the unsupervised robustness problem is a multi-objective problem. Hence, we need to look at Pareto fronts rather than scalar values. That’s why we believe that this is an interesting field to study and invite the community to propose also other unsupervised robustness measures.
>
>
> > It is not very clear the relationship between unsupervised robustness evaluation and supervised robustness evaluation. Will the unsupervised robust accuracy and supervised robust accuracy be positively correlated between them?
>
> We cannot measure “unsupervised robust accuracy” as we can measure accuracy only in a supervised setting. However, we can study if other unsupervised measures predict downstream supervised robust accuracy. For example, [here](https://ibb.co/gTKqk8z) are plots for the Targeted U-PGD with ε=0.05 and ε=0.10. They clearly show a positive correlation (0.76 and 0.78 Pearson linear correlation) between the unsupervised measures and the average certified radius of the downstream linear classifier.

---

### Official Review · Reviewer_v4fr · 2022-10-25

**Confidence:** 4
**Correctness:** 4
**Technical Novelty And Significance:** 2
**Empirical Novelty And Significance:** Not applicable
**Recommendation:** 6

**Clarity, Quality, Novelty And Reproducibility:**

The paper addresses an important problem and proposes a novel approach to it. As some of the metrics are presented in a new format, the authors need to take special care to remind the reader of the quantile version of the bounds. It is sometimes hard to follow the paper, especially the certified robustness bounds presented as quantiles in the tables. The authors do provide all the required details to reproduce the work.

**Strength And Weaknesses:**

Strengths
- The paper brings up an essential issue of the robustness of unsupervised representation models and identifies some novel task-independent metrics to measure their robustness.
- The quantile-based metrics proposed in the paper give a representation-agnostic view of robustness that allows users to compare multiple representation models.
- The authors also propose some adversarial training methods that improve the randomized smoothing-based certified robustness of the trained models.

Weaknesses
- Although the empirical attack evidence suggests that the adversarially trained representation models require more PGD iterations, this can also be explained by gradient masking. Moreover, these benefits are also not present for MOCOv3 models. Using a different attack method for the evaluation would resolve this issue.


**Summary Of The Paper:**

This paper tackles the problem of quantifying and improving the robustness of representation models in a task-agnostic fashion. Being one of the first papers to approach the problem, the authors also motivate and provide a mathematical definition for unsupervised robustness. To evaluate the unsupervised robustness of the representation models, the authors propose a generalized attack framework and use it to propose a set of metrics representation-agnostic metrics that can be used to measure the relative robustness of various representation models. Using the attacks, the authors also train adversarially robust representation models and provide a detailed empirical analysis of the relative performance of different representation models with respect to each other and adversarially trained models.

**Summary Of The Review:**

The paper points out and addresses an important problem for representation models. The paper approaches the problem by considering two possible sources of risk: breakaway risk and overlap risk. Then the authors use this to motivate various metrics for measuring the unsupervised robustness of the classifier. However, the later analysis does not use the estimated values of the two risks. So, I feel some of the empirical investigations are not well motivated. Some of the presented empirical results also need further investigation. Especially the impersonation bounds need to be recalculated with better attacks. The paper has some great ideas which could benefit the community, but I think it could use some rewriting.

---

> ### Author Response · Authors · 2022-11-11
> **Initial response to comments**
>
> > Although the empirical attack evidence suggests that the adversarially trained representation models require more PGD iterations, this can also be explained by gradient masking.
>
> Gradient masking was indeed one of the problems we were concerned about. That is why we evaluate also the certified robustness and accuracy which do not depend on adversarial attacks and cannot be tricked by gradient masking. And our adversarially trained models perform significantly better at both certified measures. In our view, this is as strong evidence as one can get that our adversarial training results in improved supervised and unsupervised robustness.
>
> > As some of the metrics are presented in a new format, the authors need to take special care to remind the reader of the quantile version of the bounds. It is sometimes hard to follow the paper, especially the certified robustness bounds presented as quantiles in the tables.
>
> Thank you for this feedback. We can see how this can indeed be hard to follow. In the revised manuscript, we will add comments in the tables denoting which measures are in percentages, which are in relative and which are in universal percentiles.
>
> Due to a mistake in our formatting, the Average Certified Radius values in the tables are presented as percentages when they refer to absolute values of l2 norms in input space. We will fix this in the revised version. Thank you for spotting it!
>
> > The paper approaches the problem by considering two possible sources of risk: breakaway risk and overlap risk. Then the authors use this to motivate various metrics for measuring the unsupervised robustness of the classifier. However, the later analysis does not use the estimated values of the two risks. So, I feel some of the empirical investigations are not well motivated.
>
> The empirical estimates of the breakaway and overlap risks are a key part of our analysis in the Results section. In fact, they are the key measure we consider in the paragraphs “There is no most robust standard model” and “Unsupervised robustness measures reveal significant differences among standard models”. Here are some excerpts:
>
> _“AMDIM is least susceptible to targeted U-PGD attacks and has the highest average certified radius but has the worst untargeted U-PGD, breakaway risk and nearest neighbor accuracy. [...] AMDIM and PixPro also have the lowest overlap risk and largest median adversarial margin. At the same time, the model with the lowest breakaway risk is SimCLR. While either AMDIM or PixPro scores the best at most measures, they both have significantly higher breakaway risk than SimCLR.”_
>
> _“AMDIM has 10.5 times higher breakaway risk than SimCLR while at the same time 2.7 times lower overlap risk than MOCOv2. [...]Additionally, AMDIM having the highest breakaway risk and lowest overlap risk indicates that unsupervised robustness is a multifaceted problem and that models should be evaluated against an array of measures.”_
>
> We do not single out the two risks in our comparison of the standard MOCOv2 and the adversarially fine-tuned versions of it as the improvement there is across every single measure. However, in Table 3 we’ve highlighted how we achieve risk values of 0.001% or less for the UNTAR and L-PGD models.
>
> We also extensively comment on the breakaway and overlap risks in the Extended Results section in the appendix.
>
> Therefore, our analysis certainly utilizes the proposed breakaway risk and overlap risk estimates.
>
> > Some of the presented empirical results also need further investigation. Especially the impersonation bounds need to be recalculated with better attacks.
>
> We feel like the reviewer might have misunderstood the impersonation rates to be calculated using supervised attacks but they are also based on the unsupervised attacks we propose. We keep the linear probe fixed and unknown to the attacker (as in the threat model in Figure 1). We then perform a targeted unsupervised attack on the encoder as described in Section C.10 in the appendix. Therefore, we can only use unsupervised adversarial attacks and, to the best of our knowledge, our work is the first to propose such. Therefore, we do not think at this moment there are better attacks to use for the impersonation evaluations.

---

### Official Review · Reviewer_rApq · 2022-10-25

**Confidence:** 4
**Correctness:** 3
**Technical Novelty And Significance:** 3
**Empirical Novelty And Significance:** 2
**Recommendation:** 5

**Clarity, Quality, Novelty And Reproducibility:**

**Clarity:** The paper is easy to understand.

**Quality:** The representation of the paper seems fine. However, I hope the authors describe which dataset is used in each table in the caption.

**Novelty:** The problem and the metric that the paper demonstrates are novel to me.

**Reproducibility:** The paper has well reproducibility which elaborates well on the details.

**Strength And Weaknesses:**

**Strength**
- This paper first tackles the problem that previous unsupervised adversarial methods only evaluated the classification task as a downstream task. To overcome such limitations, this paper proposes evaluation metrics that do not use any class labels.
- The idea and the approach to making the evaluation metric for unsupervised robustness are quite novel to me.

**Weakness**
- I am not quite sure why proposed universal quantiles and relative quantiles are able to represent the robustness of the models. Intuitively, robust models could have a relatively low ratio of universal quantiles but these metrics could represent the vulnerability that induces wrong decisions or unintended actions in downstream tasks. I think the authors could describe how we can interpret these metrics in terms of robustness.
- I think there is no big difference between U-PGD and L-PGD since the distance function is also a loss function. Further, KL divergence as a distance function could not be used in unsupervised models since there is no class probability in the unsupervised models. I might have missed but what kind of distance function is used for evaluation metric? Moreover, it seems that the L-PGD is already proposed in the previous unsupervised adversarial learning which seems to lack novelty.


**Summary Of The Paper:**

This paper proposes several evaluations of unsupervised robustness which are model agnostic and tasks agnostic. This paper examines recent unsupervised models with proposed unsupervised robustness evaluations. Specifically, the paper proposes universal quantiles for untargeted attacks and relative quantiles for targeted attacks as an evaluation metric for unsupervised robustness. The former metric represents the percentage of paired samples that are closer than the clean and adversarial pairs. The latter metric represents the ratio that the targeted attack moves the original image to the target image.

**Summary Of The Review:**

Overall, I recommend marginally below the acceptance threshold. Because whether the proposed evaluation metric could represent robustness seems unclear I hope the authors could resolve my concerns in the rebuttal.

---

> ### Author Response · Authors · 2022-11-11
> **Initial response to comments**
>
> > I am not quite sure why proposed universal quantiles and relative quantiles are able to represent the robustness of the models. Intuitively, robust models could have a relatively low ratio of universal quantiles but these metrics could represent the vulnerability that induces wrong decisions or unintended actions in downstream tasks. I think the authors could describe how we can interpret these metrics in terms of robustness.
> >
> > [...]
> >
> > Because whether the proposed evaluation metric could represent robustness seems unclear I hope the authors could resolve my concerns in the rebuttal.
>
> The definition of robustness in the supervised classification setting is simple: the model is robust at $x$ if small perturbations to $x$ do not change the predicted class. However, when we do not have a particular downstream task, defining robustness becomes much less clear-cut. Our approach to unsupervised robustness is to require the encoder to be smooth relative to small local perturbations, i.e. to map perturbed samples close to the clean sample.
>
> The intuition behind this comes from the typical use of such representations. The unsupervised encoders we consider are often converted into classifiers by adding a linear layer on top. Therefore, as the linear layer is Lipschitz, the divergences in representation space induced by a perturbation can be translated to bounds on the change of the confidence values of a downstream linear probe. More generally, keeping perturbed samples close in representation space would lead to improved robustness for smooth downstream tasks.
>
> The quantiles were proposed as means to address the problem that the value of a divergence in representation space is essentially meaningless in itself. Whether being 0.1-close in representation space is “close” or “far” depends on what are the typical divergences observed. It also differs greatly from model to model as they can have different scales of their representations. To address these issues and to enable cross-model comparison, we propose normalizing the absolute value of the divergences as percentiles of the distribution of divergences observed in representation space.
>
> We hope that this clarifies how our proposed metrics measure unsupervised robustness. We appreciate that this point is critical to the understanding of our work and will try improve the wording in the revised version of our manuscript.
>
>
> > I think there is no big difference between U-PGD and L-PGD since the distance function is also a loss function.
>
> It is correct that any distance (or divergence) function can be considered a loss function. That is the motivation behind us considering L-PGD to be a generalization of U-PGD and placing it under L-PGD in the hierarchy illustration in Figure 5. However, we would like to stress that U-PGD and L-PGD are by far not the same. There are instances of L-PGD, such as the work by Kim et al. (2020) and the L-PGD losses we use for our experiments, which are not U-PGD but are still a special case of L-PGD. Therefore, our addition of L-PGD also allows for our framework to better encompass and place prior work.
>
>
> > Further, KL divergence as a distance function could not be used in unsupervised models since there is no class probability in the unsupervised models.
>
> The KL divergence can be used in unsupervised models which do not have class probabilities if they use probabilistic representations. For example, this can be the case if one uses a VAE as the encoder. As a further example, prior work by Nguyen et al. (2022) uses the KL divergence precisely as a robustness regularizer over probabilistic representations.
>
> > I might have missed but what kind of distance function is used for evaluation metric?
>
> We use the Euclidean distance for the evaluation experiments. We provide the details in the third paragraph in the Experiments section as well as in Section C.5 in the appendix.
>
> > Moreover, it seems that the L-PGD is already proposed in the previous unsupervised adversarial learning which seems to lack novelty.
>
> To the best of our knowledge, L-PGD has not been proposed in the previous unsupervised adversarial training literature. Several individual approaches using specific choices of loss functions have been proposed. We interpret these as concrete instances of our more general approach but it would be inaccurate to claim that L-PGD has already been proposed.
>
>
>
> > The representation of the paper seems fine. However, I hope the authors describe which dataset is used in each table in the caption.
>
> We appreciate your suggestion of mentioning the dataset used in the table captions and will add them to the updated version of the manuscript.
>
>
> **References:**
>
> Minseon Kim, Jihoon Tack, and Sung Ju Hwang. Adversarial self-supervised contrastive learning. NeurIPS 2020.
>
> A. Tuan Nguyen, Ser Nam Lim, and Philip Torr. Task-agnostic robust representation learning. Preprint arXiv:2203.07596, 2022.

---

### Author Response · Authors · 2022-11-18
**Comment on the revised manuscript**

We would like to thank the reviewers for their insight and comments. We ended up implementing a number of the improvements they suggested and we believe this further strengthened our manuscript.

Reviewer HYfP suggested we evaluate a supervised encoder as well as a supervised adversarially trained encoder. We appreciated the suggestion and added a supervised adversarially trained  ResNet50 model which scored very highly on all unsupervised robustness measure that we propose. This demonstrates that our measures successfully detect models which we know to be robust in a supervised setting. We added a paragraph regarding this additional experiment to our Results section. We also added targeted and untargeted unsupervised fine-tuned ResNet50 models. The unsupervised fine-tuning improves all metrics, further strengthening our results.

Reviewer rApq recommended we mention which datasets are used in the table captions. We agree that this improves the readability of our paper so we added dataset descriptions wherever possible.

We also heeded the advice of Reviewer v4fr to designate which values are in universal or relative quantiles.

We uncovered a mistake in our calculation of the average certified radius which we have rectified. This does not affect our results.

Following some of the reviewers’ comments, we also rewrote parts of the Problem Setting section in order to improve its clarity.

Finally, a related preprint by Kim et al. (2022) was released after our initial submission. As the core method they propose is a specific instance of our L-PGD attack, we added it to Appendix A and the hierarchy of attacks in Fig. 5. We have similarly added the work by Gowal et al. (2020) which was brought to our attention. Their method is a specific instance of the untargeted U-PGD when the divergence is fixed to be the cosine similarity. We hope that the addition of these two works places our work even better in the existing literature and further illustrates how the attacks we propose unify and generalize a number of previous divergence/loss- and/or model-specific attacks.

**References:**

Sven Gowal, Po-Sen Huang, Aaron van den Oord, Timothy Mann, and Pushmeet Kohli. Self-supervised adversarial robustness for the low-label, high-data regime. In International Conference on Learning Representations, 2020.

Minseon Kim, Hyeonjeong Ha, Sooel Son, and Sung Ju Hwang. Targeted adversarial self-supervised learning. Preprint arXiv:2210.10482, 2022.

---

### Decision · Program_Chairs · 2023-01-20

**Decision:**

Reject

**Justification For Why Not Higher Score:**

See above for weaknesses

**Justification For Why Not Lower Score:**

I recommend rejection

**Metareview: Summary, Strengths And Weaknesses:**

This paper proposes several evaluations of unsupervised robustness which are model agnostic and tasks agnostic. Specifically, the paper proposes universal quantiles for untargeted attacks and relative quantiles for targeted attacks as an evaluation metric for unsupervised robustness. Reviewers generally liked the topic of the paper to study the robustness of representation learning which is an under studied problem. Although authors did a good job in addressing some of the concerns of the reviewers, there are still some remaining concerns: the issue of gradient masking should be studied more clearly in the experiments; in general reviewers had some concerns with experimental results (e.g. benefits are also not present for MOCOv3 models or different unsupervised adversarial metrics do not provide a consistent evaluation.) One reviewer suggested that the work has minor technical novelty. Given all, I think the paper needs a bit more work before being accepted.